# Polynomial-Delay MAG Listing with Novel Locally Complete Orientation Rules

Tian-Zuo Wang [1 2]   Wen-Bo Du [1 2]   Zhi-Hua Zhou [1 2]

## Abstract

A maximal ancestral graph (MAG) is widely used to characterize the causal relations among observable variables in the presence of latent variables. However, given observational data, only a partial ancestral graph representing a Markov equivalence class (MEC) of MAGs is identifiable, which generally contains uncertain causal relations. Due to the uncertainties, *MAG listing*, *i.e.*, listing all the MAGs in the MEC, is critical for many downstream tasks. In this paper, we present the first *polynomial-delay* MAG listing method, where delay refers to the time for outputting each MAG, through introducing enumerated structural knowledge in the form of *singleton background knowledge (BK)*. To incorporate such knowledge, we propose the *sound* and *locally complete* orientation rules. By recursively introducing singleton BK and applying the rules, our method can output all and only MAGs in the MEC with polynomial delay. Additionally, while the proposed novel rules enable more efficient MAG listing, for the goal of incorporating general BK, we present two counterexamples to imply that existing rules including ours, are not yet *complete*, which motivate two more rules. Experimental results validate the efficiency of the proposed MAG listing method.

## 1. Introduction

Recently, causality has gathered significant attention in artificial intelligence. In this field, a causal graph characterizing causal relations plays a vital role. On one hand, a causal graph provides a profound insight into underlying physical mechanism (Cai et al., 2018; Runge et al., 2019). On the other hand, it serves as a crucial tool for estimating causal effects (Pearl, 2009), which are essential in applications.

[1] National Key Laboratory for Novel Software Technology, Nanjing University, China [2] School of Artificial Intelligence, Nanjing University, China. Correspondence to: Zhi-Hua Zhou <zhouzh@lamda.nju.edu.cn>.

*Proceedings of the 42ⁿᵈ International Conference on Machine Learning*, Vancouver, Canada. PMLR 267, 2025. Copyright 2025 by the author(s).

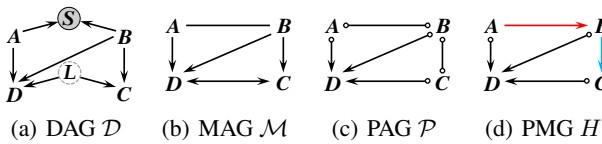

(a) DAG $\mathcal{D}$ (b) MAG $\mathcal{M}$ (c) PAG $\mathcal{P}$ (d) PMG $H$

Figure 1: Fig. 1(a): a DAG $\mathcal{D}$ including a latent confounder $L$ and a selection variable $S$. Fig. 1(b): The MAG $\mathcal{M}$ characterizing the causal relations over the observable variables in $\mathcal{D}$. Fig. 1(c): The PAG $\mathcal{P}$ representing the MEC of $\mathcal{M}$. Fig. 1(d): When orienting $A \rightarrow B$, there must be $B \rightarrow C$.

A directed acyclic graph (DAG) is a classical graphical model to characterize causal relations when all relevant variables are fully observable. In practice, however, latent variables generally exist, including latent confounders and selection variables. A latent confounder is a latent variable that influences more than one observed variable, *e.g.*, genes influence both whether a person smokes and whether this person develops lung cancer, which are typically latent. A selection variable is a latent variable influenced by multiple observed variables and plays a role in determining whether the data is collected. *E.g.*, whether a person goes to a hospital $S$ depends on both whether he is ill and whether he is alive. If the data is collected only from the hospital, it is conditioned on $S = 1$, indicating the distribution of the collected data differs from the overall population distribution.

In the presence of latent variables, maximal ancestral graph (MAG) is generally used to characterize the causal relations among observable variables (Richardson et al., 2002). Given a DAG in Fig. 1(a) with latent confounder $L$ and selection variable $S$, the corresponding MAG is as Fig. 1(b). However, existing theoretical results imply that a DAG/MAG is not identifiable from observational data (Spirtes et al., 2000), unless introducing model or functional assumptions (Shimizu et al., 2006; Zhang & Hyvärinen, 2009). In the presence of latent variables, only a Markov equivalence class (MEC) of MAGs, which encode the same conditional independence, can be identified from observational data. This MEC is represented by a partial ancestral graph (PAG), which contains uncertain structures denoted by circles (○) as Fig. 1(c).

Given that only a PAG with uncertainties is identifiable, *MAG listing*, *i.e.*, listing all the MAGs in the MEC represented by the PAG, is crucial for many downstream tasks.

For example, when uncovering causal relations within a MEC by introducing active interventions, minimizing the number of interventions is often desirable due to their high cost. The maximal entropy criterion is often employed (He & Geng, 2008; Wang et al., 2023b), requiring entropy estimation based on all graphs in the MEC, necessitating MAG listing. Additionally, in some studies on structure learning with additional structural knowledge (Kocaoglu, 2023; Gerhardus, 2024), MAG listing is also required. Despite its importance, MAG listing has received relatively limited attention in the literature, unlike DAG listing, which addresses a similar task in the absence of latent variables (Chickering, 1995; Chen et al., 2016; Wienöbst et al., 2023). Inherently, MAG listing is more challenging than DAG listing due to the existence of bi-directed edges and undirected edges.

For MAG listing, it is necessary to determine all possible orientations of the uncertain structures (circles) in the given PAG. However, directly enumerating the orientation combinations of all circles is computationally prohibitive. A more feasible approach is to recursively introduce structural knowledge and utilize orientation rules. In each round, we enumerate a part of circles. The enumerated orientation can serve as background knowledge (BK), which is then used to orient other circles by orientation rules, thereby reducing redundant enumeration. For example, consider a PAG $\mathcal{P}$ in Fig. 1(c), if we enumerate the circle at $B$ as an arrowhead on $A \circ\!\!-\!\!\circ B$, it follows $B \to C$ according to the orientation rules of Zhang (2008), making it unnecessary to enumerate the circles on the edge between $B$ and $C$. Notably, the above idea for MAG listing must be applied cautiously: When BK is introduced, the orientation rules need to be sound and complete to orient the circles that may be enumerated later, otherwise we may obtain some extra MAGs inconsistent with the PAG.[1] However, despite extensive efforts in the literature (Andrews et al., 2020; Mooij et al., 2020; Wang et al., 2023b; 2024b; Venkateswaran & Perkovic, 2024), it remains unclear what orientation rules are sound and complete for incorporating BK in the presence of latent variables.

To the best of our knowledge, Wang et al. (2024a) proposed the first MAG listing method MAGLIST that avoids exhaustive search. However, its computational complexity has not been formally analyzed. In this paper, we consider the computational complexity of MAG listing. An important metric for evaluating algorithmic complexity is *delay* (Johnson et al., 1988), which refers to the time required to produce each output. We demonstrate that MAGLIST incurs an exponential delay, because it enumerates all circles of a vertex simultaneously at each step, resulting in significant computational overhead. To address this issue, we propose a refined approach that enumerates only one circle at each step. The enumerated orientations are termed *singleton BK*. To ensure

consistency between the generated MAGs and the PAG, we introduce orientation rules that are *sound* and *locally complete* for incorporating singleton BK. Building upon these orientation rules, we develop the first polynomial-delay algorithm for MAG listing and prove that it outputs all and only the MAGs in the MEC represented by the given PAG.

Beyond MAG listing, as discussed earlier, it remains unclear what orientation rules are sound and complete for incorporating general BK in the presence of latent variables, which is crucial for solving the open problem of causal relation identification from observational data and BK (Zhang, 2008). The novel orientation rules in this paper highlight the incompleteness of existing rules. Furthermore, we provide two counterexamples to demonstrate that the existing rules, including ours, are not yet complete for incorporating BK. These counterexamples motivate two more orientation rules on incorporating BK. Our contributions are threefold.

(1) We present three novel orientation rules to incorporate singleton background knowledge which is involved in MAG listing, and prove that the rules are sound and locally complete for a broad class of graphical models.

(2) Building upon the orientation rules, we propose the first polynomial-delay MAG listing algorithm, which can output all and only the MAGs in the given MEC. Experiments validate the effectiveness and efficiency.

(3) Beyond MAG listing, we demonstrate that the existing orientation rules including ours are not yet complete for incorporating BK by two counterexamples, which further motivate two more rules on incorporating BK.

## 2. Preliminary

In this paper, we denote a vertex by uppercase letter and a set of vertices by boldface letter. The set of vertices/edges in a graph $G$ is denoted by $\mathbf{V}(G)/\mathbf{E}(G)$. For a set of vertices $\mathbf{V}' \subseteq \mathbf{V}(G)$, *the subgraph of $G$ induced by $\mathbf{V}'$* is the graph consisting with $\mathbf{V}'$ and the edges connecting $\mathbf{V}'$ in $G$, denoted by $G[\mathbf{V}']$. Denote $G[\mathbf{V}(G)\backslash\mathbf{V}']$ by $G[-\mathbf{V}']$.

A graph is *mixed* if it consists of undirected, directed, and bi-directed edges. The endpoints of an edge are referred to as *marks*. If a graph contains the three kinds of marks - arrowheads, tails, and circles, it is a *partial mixed graph (PMG)*. We use $*$ to denote a wildcard of anyone of the three marks. A structure $V_i *\!\!\to V_j \leftarrow\!\!* V_k$ is a *collider*, and we further say it is *unshielded* if $V_i$ is not adjacent to $V_k$.

For two vertices in a PMG $H$, $V_i$ is a *parent* of $V_j$ if there is $V_i \to V_j$ in $H$. $V_i \circ\!\!-\!\!\circ V_j$ is a *circle edge*. Consider a path $p = \langle V_1, V_2, \cdots, V_d \rangle$. $p$ is a *directed/circle/possible directed* path if for each edge connecting $V_i$ and $V_{i+1}$, $1 \le i \le d-1$, there is $V_i \to V_{i+1}/V_i \circ\!\!-\!\!\circ V_{i+1}$/no arrowhead at $V_i$ and

---

[1]We provide a detailed illustration in Sec. 3.1.

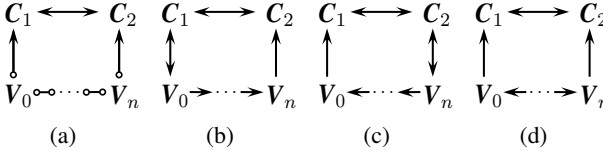

(a)  (b)  (c)  (d)

Figure 2: Fig. 2(a): a PMG with an unbridged path $V_0 \circ\!\!-\!\!\circ \cdots \circ\!\!-\!\!\circ V_n$ relative to $\{C_1, C_2\}$. Fig. 2(b)/2(c)/2(d): MAGs consistent with the PMG in Fig. 2(a).

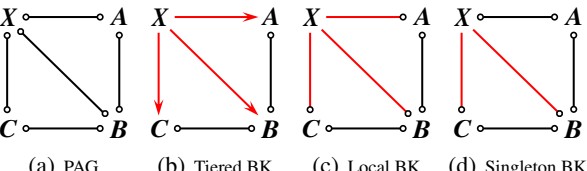

(a) PAG    (b) Tiered BK    (c) Local BK    (d) Singleton BK

Figure 3: Given a PAG in Fig. 3(a), Fig. 3(b), 3(c), and 3(d) depict tiered BK where $\{X\}$ precedes $\{A.B, C\}$, local BK regarding $X$, and singleton BK regarding $X$, respectively.

no tail at $V_{i+1}$. $p$ is *uncovered* if $\forall 1 \leq i \leq d-2$, $V_i$ is not adjacent to $V_{i+2}$. $p$ is *minimal* if any two non-consecutive vertices are not adjacent. $V_i$ is an *ancestor/possible ancestor* of $V_j$ if there is a directed/possible directed path from $V_i$ to $V_j$ or $V_i = V_j$. *Descendant/possible descendant* is defined similarly. Denote the set of vertices that are parents/ancestors/descendants/possible descendants of $V$ in $H$ by $\mathrm{Pa}(V, H)/\mathrm{Anc}(V, H)/\mathrm{De}(V, H)/\mathrm{PossDe}(V, H)$. $V_i \in \mathrm{Anc}(\mathbf{V}, H)$ if there is $V \in \mathbf{V}$ such that $V_i \in \mathrm{Anc}(V, H)$.

We guide readers to Richardson et al. (2002) for *ancestral* and *maximal* property. If a mixed graph fulfills ancestral and maximal property, it is a *maximal ancestral graph (MAG)*, denoted by $\mathcal{M}$. A *partial ancestral graph (PAG)*, denoted by $\mathcal{P}$, represents a MEC of MAGs. Based on $\mathcal{P}$, we could obtain a PMG $H$ by orienting some circles as arrowheads or tails, a MAG $\mathcal{M}$ *is consistent with* $H$ if it belongs to the MEC represented by $\mathcal{P}$ and has the non-circle marks in $H$. Given background knowledge (BK), $\mathcal{M}$ is consistent with BK if $\mathcal{M}$ has the marks contained in the BK.

**Definition 1** (Unbridged path relative to $\mathbf{V}'$; (Wang et al., 2024b)). Let $\mathbf{V}'$ be a subset of vertices in a PMG $H$. A path $p : V_0 \circ\!\!-\!\!\circ \cdots \circ\!\!-\!\!\circ V_n$ is an unbridged path relative to $\mathbf{V}'$ if there is $C_1, C_2 \in \mathbf{V}'$ such that there is a minimal path $C_1 \leftarrow\!\!\circ V_0 \circ\!\!-\!\!\circ V_1 \circ\!\!-\!\!\circ \cdots \circ\!\!-\!\!\circ V_n \circ\!\!\rightarrow C_2$ except for an edge $C_1 \leftrightarrow C_2$ in $H$.

We present *unbridged path* in Def. 1, with an example in Fig. 2(a). Intuitively, given a PMG $H$, unbridged path relative to $\mathbf{V}'$ describes a case that in any MAG $\mathcal{M}$ consistent with $H$, each vertex $V$ in the unbridged path is always an ancestor of $\mathbf{V}'$. See Fig. 2, in any MAG consistent with the PMG in Fig. 2(a), since there cannot be new unshielded colliders introducing additional conditional independence, the MAG can only be as Fig. 2(b), 2(c), or 2(d), where each $V_i, 0 \leq i \leq n$ is always an ancestor of either $C_1$ or $C_2$.

To learn a PAG with observational data, Ali et al. (2005a); Zhang (2008) presented some orientation rules. Further, rules for incorporating specific types of BK, such as *tiered BK* (Andrews et al., 2020) and *local BK* (Wang et al., 2023b), were proposed, along with others (Wang et al., 2024b; Venkateswaran & Perkovic, 2024). These rules $\mathcal{R}_1 - \mathcal{R}_{13}$ are provided in App. A. Specifically, tiered BK organizes

vertices into ordered components, where relations between components are known, while those within each component remain uncertain. In contrast, local BK provides *full* structural knowledge regarding several vertices. Consider a PAG in Fig. 3(a). Fig. 3(b) and Fig. 3(c) depict tiered BK where $\{X\}$ precedes $\{A.B, C\}$ and local BK regarding $X$.

Finally, we introduce a broad class of PMGs, *PMG compatible with local transformation*, in Def. 4 in App. B. Briefly speaking, if a PMG fulfills *chordal*, *balance*, *complete*, and *constructive* conditions, then it is a PMG compatible with local transformation. See App. B for details. Evidently, the PAG fulfills the four conditions, thus PAG is a special case of PMG compatible with local transformation.

## 3. The Proposed Method

In Sec. 3.1, we revisit existing MAG listing approach, and identify the bottleneck leading to redundant computational complexity. To overcome the limitation, we investigate *singleton BK*, a critical component in MAG listing. In Sec. 3.2, we propose three novel orientation rules to incorporate singleton BK. Building on these rules, we present a polynomial-delay MAG listing algorithm in Sec. 3.3. Discussions are presented in Sec. 3.4. All the proofs are given in App. D.

### 3.1. Revisiting MAGLIST

In this part, we revisit the MAG listing method MAGLIST proposed by Wang et al. (2024a) and briefly explain the reasons behind its exponential delay, which motivates the development of our polynomial-delay method.

Given a PAG $\mathcal{P}$ learned from observational data, MAGLIST obtains all MAGs consistent with $\mathcal{P}$ by recursively enumerating the circles of each vertex. A copy of the realization process of MAGLIST is shown in App. C. Briefly speaking, in each round, MAGLIST selects a vertex with circles, denoted by $X$, then (a) enumerates all the circles at $X$ as non-circles; (b) updates the graph based on the enumerations. In step (a), each enumerated non-circle configuration at $X$ dictates a local structure of $X$. Wang et al. (2024a) presented the graphical condition to determine whether it is valid, *i.e.*, whether there is a MAG consistent with $\mathcal{P}$ and

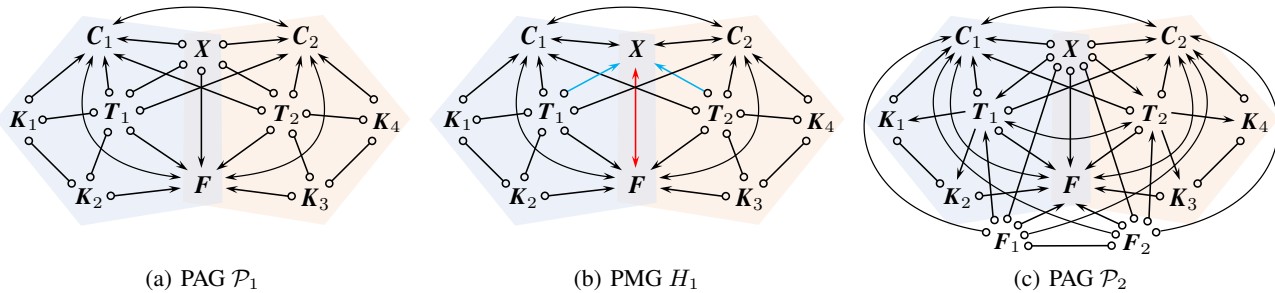

(a) PAG $\mathcal{P}_1$        (b) PMG $H_1$        (c) PAG $\mathcal{P}_2$

Figure 4: Two examples to show the incompleteness of the existing orientation rules. A PAG $\mathcal{P}_1$ is shown in Fig. 4(a) where there must be $X \to F$ when introducing BK $C_1 \ast\!\!\to X \leftarrow\!\!\ast C_2$. For contradiction, suppose there is $X \leftrightarrow F$, as shown in Fig. 4(b), the blue edges $T_1 \circ\!\!\to X \leftarrow\!\!\circ T_2$ are oriented by $\mathcal{R}_{12}$, which form an unshielded collider. Another PAG $\mathcal{P}_2$ is shown in Fig. 4(c), where there is also $X \to F$ but is not identifiable by existing rules when introducing BK $C_1 \ast\!\!\to X \leftarrow\!\!\ast C_2$.

the local structure. Each enumerated valid local structure introduces local BK regarding $X$. Hence, the existing orientation rules for incorporating local BK can soundly and completely update the graph in step (b). It is proved that the graph obtained after each round is a PMG compatible with local transformation, enabling the process to be recursively executed for every vertex. Let $d$ denote the number of vertices. In step (a), the enumeration of the circles at $X$ has a complexity of $\mathcal{O}(2^d)$, while the number of valid local structures could be $\mathcal{O}(d)$, leading to an exponential delay.[2]

Hence, the simultaneous enumeration of all the circles at $X$ is the main bottleneck for the undesired complexity. To develop a polynomial-delay method, it is essential to eliminate the redundant enumeration in step (a). We focus on this step. A direct idea is to enumerate each circle at $X$ one by one, instead of simultaneously. After transforming a circle, we apply the orientation rules to update the graph, thereby avoiding redundant enumeration. For example, given a PMG in Fig. 1(c), consider enumerating the circles at $B$. If we first orient $A \circ\!\!\to B$, then $B \to C$ follows by the rules as Fig. 1(d), avoiding enumerating the circle at $B$ on $B \circ\!\!-\!\!\circ C$.

However, the above idea cannot be applied directly. It might not only introduce additional computational overhead but, more importantly, result in outputting MAGs inconsistent with $\mathcal{P}$. This is because when we enumerate only one or a proper subset of circles at $X$, the existing orientation rules are not *locally complete* to update the graph, *i.e.*, some circles at $X$ that *should have been identified* as tails or arrowheads might remain unidentified. In such cases, we will obtain extra MAGs by subsequent enumerations. See Fig. 1(c) for an example. Consider enumerating the circle at $B$ on $A \circ\!\!-\!\!\circ B$ as an arrowhead. Since the rules are not locally complete, suppose the circle at $B$ on $B \circ\!\!-\!\!\circ C$ is not identified, which should have been identified by $\mathcal{R}_1$ given $A \circ\!\!\to B$. In this case, we would subsequently enumerate the circle

at $B$ on $B \circ\!\!-\!\!\circ C$ as an arrowhead, leading to a PMG with $A \circ\!\!\to B \leftarrow\!\!\circ C$. All the MAGs obtained from this PMG are inconsistent with $\mathcal{P}$, as the additional unshielded colliders relative to $\mathcal{P}$ indicate extra conditional independencies.

As discussed above, when enumerating each circle at $X$ one by one, it is vital to establish sound and locally complete orientation rules to transform the circles at $X$. Each transformation at $X$ can be viewed as a type of BK. Hence, we study orientation rules to incorporate such BK in Sec. 3.2.

### 3.2. The proposed rules to incorporate singleton BK

In this part, we propose the orientation rules for incorporating *singleton BK* into a PMG, where singleton BK refers to the additional structural information regarding a singleton vertex. Using the rules, during MAG listing process, we can update the graph with the enumerations of circles regarding the selected variable. Since a PAG can be identified from observational data (Zhang, 2008), we make a convention that when we say PMG, it refers to a PAG or a PMG obtained based on a PAG, *i.e.*, the PMG at least contains the structural information in a PAG. We assume BK is correct, ensuring the existence of MAGs consistent with $\mathcal{P}$ and BK.

**Definition 2** (singleton BK). Given a PMG $H$ and BK, BK is singleton if the BK only directly indicates the orientation of some circles regarding one vertex.

Singleton BK implies only the orientation of circles regarding one vertex, as shown in Fig. 3(d) where BK regarding $X$ is introduced. Note that singleton BK does not necessarily indicate the orientation of the circles regarding all the vertices of the vertex. It is the main distinction between singleton and local BK.

As discussed earlier, given singleton BK, an important question for MAG listing is whether the existing orientation rules are sound and locally complete for incorporating such BK. We demonstrate the incompleteness by presenting two counterexamples in Fig. 4. Consider a PAG in Fig. 4(a) and singleton BK regarding $X$ as $C_1 \ast\!\!\to X \leftarrow\!\!\ast C_2$. Using the

---

[2]We provide an example to demonstrate that MAGLIST could incur an exponential delay in Sec. 3.4.

existing rules, no new orientations are triggered; however, $X \to F$ should be identified. We illustrate it by contradiction. Suppose a PMG $H_1$ as Fig. 4(b) satisfies the singleton BK but includes $X \leftrightarrow F$. For the left shaded region in $H_1$, $T_1 \circ\!\!\to X$ is oriented by $\mathcal{R}_{12}$ (see App. A for $\mathcal{R}_{12}$), as there is an unbridged path $K_1 \circ\!\!-\!\!\circ K_2$ relative to $\{C_1, F\}$. Similarly, for the right shaded region, $T_2 \circ\!\!\to X$ is oriented. Consequently, a new unshielded collider $T_1 \circ\!\!\to X \leftarrow\!\circ T_2$ is formed. However, introducing new unshielded colliders relative to $\mathcal{P}$ is *forbidden*, as it implies conditional independence that differs from $\mathcal{P}$ such that any MAGs with this structure are inconsistent with $\mathcal{P}$. If we examine why $X \leftrightarrow F$ is impossible in Fig. 4(b) given the singleton BK regarding $X$, orienting $X \circ\!\!\to F$ as $X \to F$ will lead to new arrowheads at $X$ by $\mathcal{R}_{12}$, forming new unshielded colliders that are forbidden. Beyond $\mathcal{R}_{12}$, similar issues may arise with $\mathcal{R}_2$ and $\mathcal{R}_{11}$. To formalize the introduction of new arrowheads by $\mathcal{R}_2, \mathcal{R}_{11}, \mathcal{R}_{12}$, we define *prior to* in Def. 3.

**Definition 3.** Consider a PMG $H$. For two vertices $A, B$ with $A \ast\!\!-\!\!\circ X \circ\!\!-\!\!\ast B$, we say that $A$ is prior to $B$ relative to $X$ if there exists a set of vertices $F_0(= A), F_1, \cdots, F_t(= B), t \geq 1$ such that for any $0 \leq i \leq t-1$, there is $F_i \ast\!\!-\!\!\circ X$, and one of the three following conditions holds:

(1) there is an edge $F_i \to F_{i+1}$ in $H$,

(2) there is an uncovered possible directed path $\langle X, F_i, \cdots, M \rangle$ and an edge $M \to F_{i+1}$ in $H$,

(3) there is an unbridged path $\langle K_1, \cdots, K_m \rangle$ relative to $\mathbf{S}_{F_{i+1}}$ in $H[-\mathbf{S}_{F_{i+1}}]$, and for every vertex $K_j \in \{K_1, \cdots, K_m\}$, there exists an uncovered possible directed path $\langle X, F_i, \cdots, K_j \rangle$ $(F_i \neq K_j)$, where $\mathbf{S}_{F_{i+1}} = \{V \in \mathbf{V}(H) | V \ast\!\!\to X \text{ in } H\} \cup \{F_{i+1}\}$.

Intuitively, for vertices $F_i$ and $F_{i+1}$ in Def. 3 fulfilling one of the three conditions, if there is $X \leftrightarrow F_{i+1}$, then $X \circ\!\!-\!\!\ast F_i$ is transformed to $X \leftarrow\!\!\ast F_i$ by $\mathcal{R}_2/\mathcal{R}_{11}/\mathcal{R}_{12}$, of which the corresponding conditions are shown in Fig. 5. Hence, if $A$ is prior to $B$ relative to $X$ as Def. 3, when $X \circ\!\!-\!\!\ast B$ is transformed to $X \leftarrow\!\!\ast B$, $X \circ\!\!-\!\!\ast A$ will be finally transformed to $X \leftarrow\!\!\ast A$ by repeatedly triggering $\mathcal{R}_2, \mathcal{R}_{11}, \mathcal{R}_{12}$.

Based on Def. 3, we present $\mathcal{R}_{14}$, which is inspired by the counterexamples. Intuitively, if $T_1$ and $T_2$ are prior to $B$ relative to $X$ in $\mathcal{R}_{14}$, orienting $X \leftarrow\!\!\ast B$ implies $X \leftarrow\!\!\ast T_1$ and $X \leftarrow\!\!\ast T_2$. In this case, given the conditions in $\mathcal{R}_{14}$ that (a) there is an unbridged path $F_1 \circ\!\!-\!\!\circ \cdots \circ\!\!-\!\!\circ F_k$ relative to $\{T_1, T_2\}$ such that $X \circ\!\!-\!\!\ast F_i, \forall F_{i,1\leq i\leq k}$, or (b) $T_1$ is not adjacent to $T_2$, there must be new unshielded colliders introduced, which is forbidden in the process of incorporating BK. The example for case (b) in $\mathcal{R}_{14}$ has been provided in Fig. 4(a). For case (a), we give another example in Fig. 4(c). In this example, $T_1$ is adjacent to $T_2$, but there is an unbridged path $F_1 \circ\!\!-\!\!\circ F_2$ relative to $\{T_1, T_2\}$. In this case, there is $X \to F$ oriented as well.

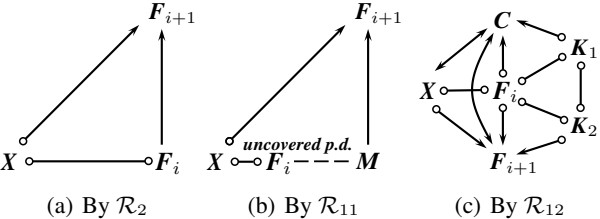

Figure 5: Examples of that $V_i$ is prior to $V_{i+1}$ relative to $X$. If we orient $X \circ\!\!-\!\!\ast F_{i+1}$ as $X \leftarrow\!\!\ast F_{i+1}$ in Fig. 5(a)/5(b)/5(c). $X \circ\!\!-\!\!\ast F_i$ will be oriented as $X \leftarrow\!\!\ast F_i$ by $\mathcal{R}_2/\mathcal{R}_{11}/\mathcal{R}_{12}$.

$\mathcal{R}_{14}$ Suppose an edge $X \circ\!\!\to B$ in a PMG $H$, if there is $T_1 \ast\!\!-\!\!\circ X \circ\!\!-\!\!\ast T_2$ where both $T_1$ and $T_2$ are prior to $B$ relative to $X$, and there is either (a) an unbridged path $F_1 \circ\!\!-\!\!\circ \cdots \circ\!\!-\!\!\circ F_k$ relative to $\{T_1, T_2\}$ such that $X \circ\!\!-\!\!\ast F_i$, $\forall F_{i,1\leq i\leq k}$, or (b) $T_1$ is not adjacent to $T_2$, then orient $X \circ\!\!\to B$ as $X \to B$.

$\mathcal{R}_{15}$ If $A \ast\!\!-\!\!\ast B \to R$, and $A \circ\!\!\to R$, orient $A \circ\!\!\to R$ as $A \to R$. (generalization of $\mathcal{R}_8$)

$\mathcal{R}_{16}$ If $A \ast\!\!\to B \multimap R$ or $A \ast\!\!\to B \circ\!\!-\!\!R$, then orient $B \multimap R$ or $B \circ\!\!-\!\!R$ as $B \to R$ or $B \leftarrow R$.

Additionally, two additional rules $\mathcal{R}_{15}, \mathcal{R}_{16}$ are introduced to accommodate the presence of undirected edges. Hereafter, the term *proposed orientation rules* refers to $\mathcal{R}_1 - \mathcal{R}_{16}$. Prop. 1 implies that $\mathcal{R}_{14} - \mathcal{R}_{16}$ are sound for incorporating BK. The soundness of the remaining rules has been demonstrated in previous studies (Ali et al., 2005b; Zhang, 2008; Wang et al., 2024b; Venkateswaran & Perkovic, 2024).

**Proposition 1.** $\mathcal{R}_{14} - \mathcal{R}_{16}$ *are sound for incorporating background knowledge to a PMG.*

In fact, $\mathcal{R}_8$ and $\mathcal{R}_{10}$ are special cases of $\mathcal{R}_{15}$ and $\mathcal{R}_{14}$, respectively. Since $\mathcal{R}_8$ and $\mathcal{R}_{10}$ are canonical rules, we introduce new rules to extend their scope. Next, we present our main result in Thm. 1, demonstrating that the proposed rules are *locally complete* for incorporating singleton BK into any PMGs compatible with local transformation.

**Theorem 1** (Locally complete). Consider a vertex $X$ in a PMG $\mathbb{M}$ compatible with local transformation, the proposed orientation rules $\mathcal{R}_1 - \mathcal{R}_{16}$ are locally complete for introducing singleton BK regarding $X$ into $\mathbb{M}$. That is, in the PMG $H$ obtained from $\mathbb{M}$ using the singleton BK and orientation rules, if there is an edge $X \circ\!\!-\!\!\ast F_1$ in $H$, there must exist two MAGs $\mathcal{M}_1, \mathcal{M}_2$ consistent with $\mathbb{M}$ and singleton BK, such that there is $X \leftarrow\!\!\ast F_1$ in $\mathcal{M}_1$ and $X \to\!\!\ast F_1$ in $\mathcal{M}_2$.

Despite being *locally complete*, the proposed rules are not necessarily *complete* for incorporating singleton BK. Local complete property ensures that by applying the orientation rules, we can identify all marks at $X$ that are identifiable, but it imposes no restrictions on the identification of marks

at other vertices. In Sec. 3.3, we demonstrate that locally complete rules suffice to develop a polynomial-delay MAG listing algorithm. We discuss the completeness in Sec. 4.

**The complexity of implementing the rules.** In this paper, we do not touch much on the computational complexity to implement the rules. We provide a rough analysis. Suppose a PMG has $d$ vertices and $m$ edges. For $\mathcal{R}_{11}$, there are $\mathcal{O}(m)$ edges $A \circ\!\!-\!\!* B$ that can be possibly transformed. Finding $\mathbf{S}_A$ and every vertex that can be $K_i$ requires $\mathcal{O}(d)$ and $\mathcal{O}(m)$, respectively. The total complexity is $\mathcal{O}(m^2 d)$. For $\mathcal{R}_{12}$, there are $\mathcal{O}(d)$ variable where circles are transformed. Finding each vertex $K_i$ takes $\mathcal{O}(m)$, and detecting the existence of unbridged paths requires $\mathcal{O}(m^2)$. Hence the total complexity is $\mathcal{O}(m^3 d)$. For $\mathcal{R}_{14}$, For each variable $X$, there are $\mathcal{O}(d^2)$ pairs of $V_i, V_j$ such that $V_i *\!\!-\!\!\circ X \circ\!\!-\!\!* V_j$, and detecting whether $V_i$ is prior to $V_j$ (or vice versa) relative to $X$ takes $\mathcal{O}(m^3 d)$. Additionally, there are $\mathcal{O}(d)$ variables $X$ such that $X \circ \to V$ is possibly oriented by $R_{14}$. For each such $X$, since we already know which variables are prior to $V$ relative to $X$, enumerating each pair $T_1, T_2$ prior to $V$ takes $O(d^2)$, and detecting conditions (a) and (b) in $R_{14}$ takes $\mathcal{O}(m)$. Hence, the total complexity is $\mathcal{O}(d) * \mathcal{O}(d^2) * \mathcal{O}(m^3 d) + \mathcal{O}(d) * \mathcal{O}(d^2) * \mathcal{O}(m) = \mathcal{O}(m^3 d^4)$. For $\mathcal{R}_{16}$, the complexity is $\mathcal{O}(m^2)$. For $\mathcal{R}_{17}$, there are $\mathcal{O}(m)$ edges that can be $D *\!\!\to C$. To find the vertices $A, B$, the complexity is $\mathcal{O}(d^2)$. Hence the total complexity is $\mathcal{O}(m d^2)$. The complexity of implementing $\mathcal{R}_1 - \mathcal{R}_{10}$ is less than $\mathcal{O}(m^3 d^4)$. Thus, the overall complexity of implementing the rules is $\mathcal{O}(m^3 d^4)$. The complexity is significantly higher than that of $\mathcal{R}_1 - \mathcal{R}_{10}$, which implies the intrinsic difficulty of incorporating BK with latent variables.

## 3.3. Polynomial-delay MAG listing

In this part, based on the rules for incorporating singleton BK, we present our MAG listing method MAGLIST-POLY.

The algorithm is detailed in Alg. 1. Starting from a PAG $\mathcal{P}$ learned from observational data, the method operates through two nested loops implemented as recursive functions. In the outer loop (Function ORIENTGRAPH), each iteration selects a vertex with circles, denoted as $X$, and invokes the inner loop. In the inner loop (Function LOCAL-TRANSFORM), each iteration enumerates a circle at $X$ as an arrowhead and a tail. The resulting two graphs are then updated using the proposed orientation rules. This process continues recursively until there are no circles at $X$. The obtained graphs are passed back to the outer loop. The outer loop stops when there is no vertex with circles. Notably, all the graphs obtained on Line 17 are PMGs compatible with local transformation.[3] Therefore, in each round of outer loop, the input graph is a PMG compatible with local transformation, in which case the condition of Thm. 1 holds.

---

[3]We detail it in the proof of Thm. 2.

---

**Algorithm 1** MAGLIST-POLY

**Require:** A PAG $\mathcal{P}$
1: $\mathcal{S} = \emptyset$ ▷ Record all the MAGs consistent with $\mathcal{P}$
2: ORIENTGRAPH($\mathcal{P}$)
3: **function** ORIENTGRAPH($\mathbb{M}, \mathcal{S}$)
4:     **if** there are no circles in $\mathbb{M}$ **then** ▷ $\mathbb{M}$ is a MAG
5:         $\mathcal{S} \leftarrow \mathcal{S} \cup \{\mathbb{M}\}$
6:     **else**
7:         Select a variable $X$ where there are circles in $\mathbb{M}$
8:         $\mathcal{I} = \emptyset$ ▷ Record all the PMGs without circles at $X$ obtained from $\mathbb{M}$
9:         LOCALTRANSFORM($\mathbb{M}, X$)
10:         **for** $\mathbb{M}'$ in $\mathcal{I}$ **do**
11:             ORIENTGRAPH($\mathbb{M}', \mathcal{S}$)
12:         **end for**
13:     **end if**
14: **end function**
15: **function** LOCALTRANSFORM($H, X$)
16:     **if** there are no circles at $X$ in $H$ **then**
17:         $\mathcal{I} \leftarrow \mathcal{I} \cup \{H\}$
18:     **else**
19:         Select a vertex $V$ with an edge $X \circ\!\!-\!\!* V$ in $H$
20:         Obtain $H'$ from $H$ by orienting $X \to\!\!* V$ and using the proposed orientation rules
21:         LOCALTRANSFORM($H', X$)
22:         Obtain $H''$ from $H$ by orienting $X \leftarrow\!\!* V$ and using the proposed orientation rules
23:         LOCALTRANSFORM($H'', X$)
24:     **end if**
25: **end function**
**Ensure:** $\mathcal{S}$

---

The realization process of an inner loop is shown in Fig. 6 using a search tree. Suppose $X$ is the selected vertex where circles are being transformed. The root is a PMG compatible with local transformation. In the first round, we enumerate the circle at $X$ on $X \circ \to A$. Enumerating it as an arrowhead and a tail yields two graphs, shown in the first and fourth graphs in the second layer. These graphs are further updated using the orientation rules, resulting in the two additional graphs in this layer. In the second round, we enumerate the circle at $X$ on $X \circ\!\!-\!\!\circ B$. Due to space limit, only the graphs obtained after both enumerating a circle at $X$ and using the rules are displayed in the third and fourth layers. The shaded graphs denote the PMGs without circles at $X$, which are passed back to outer loop. For the PMGs with circles at $X$, we continue enumerating. Through the search process, we obtain all valid orientations of circles at $X$.

Finally, for MAGLIST-POLY, Thm. 2 guarantees that it can find all and only the MAGs consistent with $\mathcal{P}$, and Thm. 3 ensures the polynomial delay. These results establish the theoretical effectiveness and efficiency of MAGLIST-POLY.

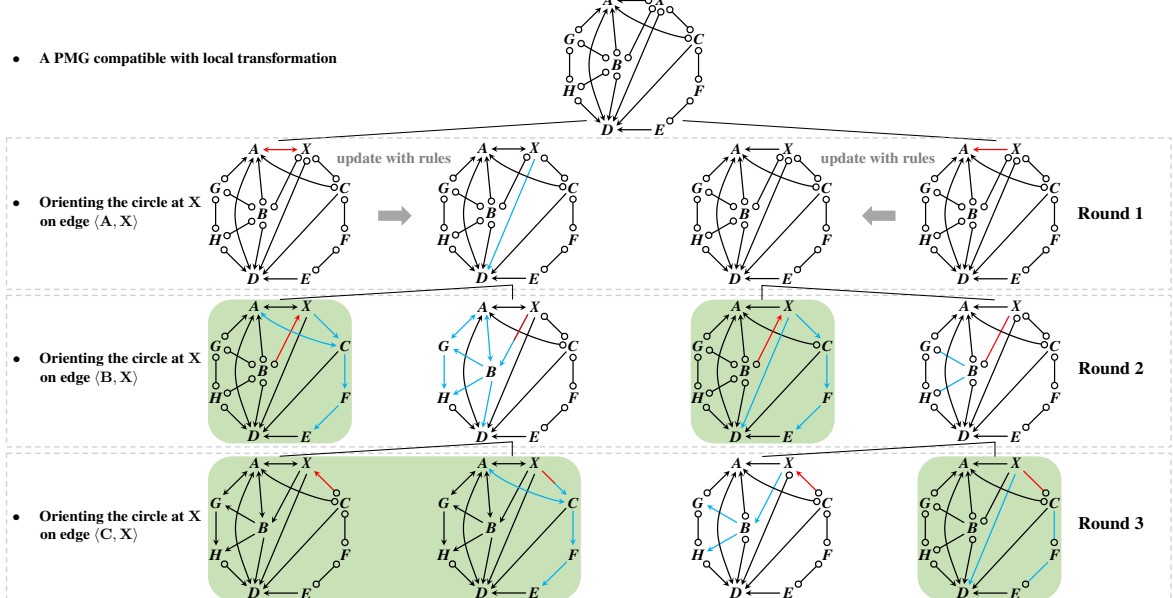

Figure 6: A realization process of Line 9 of Alg. 1, which transforms the circles at $X$ given a PMG compatible with local transformation. In the first/second/third round, we transform the circle at $X$ on the edge between $X$ and $A/B/C$. After each transformation, we further use the rules to update. Shaded graphs denote the PMGs without circles at $X$, where the inner loop stops and the outer loop is executed to select the next vertex with circles. Due to space limit, we show the circle transformation at $X$ (colored by red) and the update with rules (colored by blue) respectively only for the first round.

**Theorem 2.** Algorithm 1 is valid to list all and only the MAGs consistent with $\mathcal{P}$.

**Theorem 3.** Given a PAG $\mathcal{P}$ with $d$ vertices and $m$ edges, suppose there are $N$ MAGs consistent with $\mathcal{P}$. Denote the complexity of implementing Alg. 1 for $\mathcal{P}$ by $T(N, d)$. Then $T(N, d)/N \leq \mathcal{O}(m^3 d^4)$.

### 3.4. Discussions on MAG listing

According to the results above, we have demonstrated that using sound and locally complete rules for incorporating singleton BK suffices to develop a polynomial-delay MAG listing algorithm. Next, we present an example to illustrate that MAGLIST proposed by Wang et al. (2024a) can incur exponential-delay. Consider a PAG $\mathcal{P}$ in Fig. 7. When MAGLIST is applied, the method selects $X$ and enumerates the circles at $X$. In this case, the method will enumerate $2^d$ possible orientations, resulting in a total complexity of at least $\mathcal{O}(2^d)$. Next, consider the number of MAGs consistent with $\mathcal{P}$. Since no additional unshielded colliders can appear in these MAGs, the number of valid local structures of $X$ is $d + 1$. The delay is $\mathcal{O}(2^d/d)$, indicating exponential delay.

Compared to MAGLIST, our method achieves polynomial delay by transforming one circle at a time. This raises a natural question: is the locally complete property of the rules necessary for achieving polynomial delay? If the rules are not locally complete, then for each local transformation obtained from transforming the circles one by one at a vari-

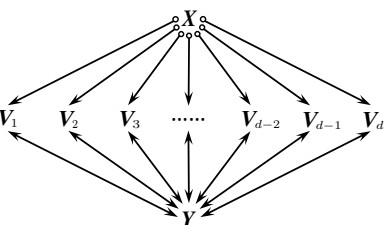

Figure 7: An example to show that MAGLIST proposed by Wang et al. (2024a) could incur an exponential delay.

able, we need to perform an additional validity check using Lemma 2 in App. D, which is costly yet effective for MAG listing. In such cases, extra time is spent detecting and discarding invalid transformations, which could be avoided by using locally complete rules. Now, it is unclear whether this extra computation leads to a delay exceeding polynomial time. The main challenge lies in that the number of MAGs consistent with a given PAG is unknown, making it difficult to analyze the delay introduced by the extra computations.

Regarding the calculation of the number of MAGs consistent with a given PAG, a related task in the DAG setting, known as *DAG counting*, aims to determine the number of DAGs within a MEC. This problem has been extensively studied (He et al., 2015; Ganian et al., 2022), and can be solved with a complexity of $\mathcal{O}(d^4)$ (Wienöbst et al., 2023). However, for the analogous task of *MAG counting*, to the best of our knowledge, there are no relevant studies. There are two

main challenges for generalizing DAG counting to MAG counting: (1) Bi-directed edges in ancestral graphs prevent using an *order* to describe uncertain edge orientations, and it is unclear what mathematical tool could replace order in this context; (2) The orientation of $\circ\!\rightarrow$ connecting different circle components can affect the orientation within a circle component. For example, consider $A \circ\!\!-\!\!\circ B \circ\!\rightarrow C \leftarrow\!\!\circ A$. If there is $B \leftrightarrow C \leftarrow A$, there must be $A\ast\!\!\rightarrow B$, illustrating that the orientations within a component are not independent of those outside. Determining the size of the MEC in the presence of latent variables remains an open problem.

## 4. Additional Rules for Incorporating BK

Beyond MAG listing, establishing sound and complete rules for incorporating BK into an MEC is a fundamental problem in causal inference (Zhang, 2008). It is essential for determining which causal relations can be identified from BK and observational data, with BK derived from interventions or human expertise. In the absence of latent variables, the canonical Meek rules are sound and complete (Meek, 1995); but it remains unresolved in the presence of latent variables.

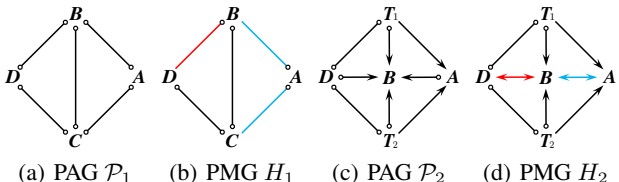

(a) PAG $\mathcal{P}_1$    (b) PMG $H_1$    (c) PAG $\mathcal{P}_2$    (d) PMG $H_2$

Figure 8: Examples to show the incompleteness of the proposed rules. Given PAG $\mathcal{P}_1$ in Fig. 8(a), when we add BK $D \circ\!\!-\!\!\circ B$ as Fig. 8(b), the proposed rules ($\mathcal{R}_1 - \mathcal{R}_{16}$) cannot identify $C \circ\!\!-\!\!\circ A$. Given $\mathcal{P}_2$ in Fig. 8(c), when we add BK $D \leftrightarrow B$ as Fig. 8(d), the rules cannot identify $A \leftrightarrow B$.

In Sec. 3.2, to develop a polynomial-delay MAG listing method, we propose novel orientation rules, and prove that they are sound and locally complete for incorporating singleton BK into a PMG compatible with local transformation. The additional rules $\mathcal{R}_{14} - \mathcal{R}_{16}$ reveal the incompleteness of the existing rules, as singleton BK is a specific type of BK. A direct question arises as to whether the rules in this paper achieve completeness for incorporating BK. To address this, we extend our investigation to the completeness of these rules beyond the MAG listing algorithm. Unfortunately, the rules are not yet complete. We present two examples in Fig. 9 implying the incompleteness, highlighting the need for further study. Motivated by the examples, we present two additional rules, whose soundness is ensured by Prop. 2.

$\mathcal{R}_{17}$ Suppose $D \ast\!\!-\!\!\ast C \ast\!\!-\!\!\ast A \ast\!\!-\!\!\ast D$ and $C \rightarrow\!\!\ast B \ast\!\!-\!\!\circ A$ in a PMG $H$, if $D$ is not adjacent to $B$, then orient $A \circ\!\!-\!\!\ast B$ as $A \ast\!\!-\!\!B$.

$\mathcal{R}_{18}$ If $A\circ\!\rightarrow B \leftrightarrow D$, there are two minimal possible directed paths $\langle D, T_1, \cdots, A\rangle$ and $\langle D, T_2, \cdots, A\rangle$, if (a) there is an unbridged path $F_1 \circ\!\!-\!\!\circ \cdots \circ\!\!-\!\!\circ F_k$ relative to $\{T_1, T_2\}$ such that $D \circ\!\!-\!\!\ast F_i, \forall F_{i,1\leq i\leq k}$, or (b) $T_1$ is not adjacent to $T_2$, then orient $A\circ\!\rightarrow B$ as $A \leftrightarrow B$.

**Proposition 2.** $\mathcal{R}_{17}, \mathcal{R}_{18}$ *are sound for incorporating BK.*

## 5. Related Works

Dealing with latent variables is vital in causal inference. One line is to recover latent variables and discover causal relations among observed and latent variables (Xie et al., 2022; Li et al.). Another line focuses on identifying causal relations or causal effects among observed variables without explicitly recovering the latent variables (Tian & Pearl, 2002; Miao et al., 2018; Lee et al., 2019; Jung et al., 2023). For example, Wang et al. (2023a) proposed a method for determining the set of possible causal effect of a variable $X$ on a variable $Y$ given a MEC with latent confounders. Wang & Miao (2024) presented sound average causal effect identification result with unmeasured confounders under the light-tailedness assumption. Ancestral graphs are such a kind of graphical model to characterize causal relations among observed variables (Richardson et al., 2002), and many studies are developed based on ancestral graphs (Cheng et al., 2022; Hu & Evans, 2024; Park et al., 2025).

Causality attracts extensive attention in decision tasks (Lee & Bareinboim, 2018; Ruan et al., 2024). Recently, a decision problem called *avoiding undesired future (AUF)* is studied, *i.e.*, if an ML model predicts an undesired event, how to find effective actions to prevent it. Considering that causal relations are often hard to identify in practice and, even when identifiable, may not be helpful if they are unactionable, Zhou (2023) proposed the concept of *influence* relation, which lies between correlation and causation, to address AUF problems (Zhou, 2022). Based on graphical models characterizing influence relations, some efforts are made for AUF problems (Qin et al., 2023; Du et al., 2024).

## 6. Experiments

In this section, we evaluate the effectiveness and efficiency of the proposed MAG listing algorithm. Our method, MAGLIST-POLY, is compared with MAGLIST and BRUTE-FORCE, which are presented by Wang et al. (2024a).

We follow the experimental setup of (Wang et al., 2024a), focusing on MAG listing under varying parameters. There are two parameters: the number of vertices $d$ and graph density $\rho$ - the probability of an edge between two vertices. For each combination of $d$ and $\rho$, we generate 100 Erdös-Rényi graphs as DAGs. For each DAG, three vertices are randomly selected as latent variables to generate a MAG $\mathcal{M}$ characterizing the causal relations among the observed variables.

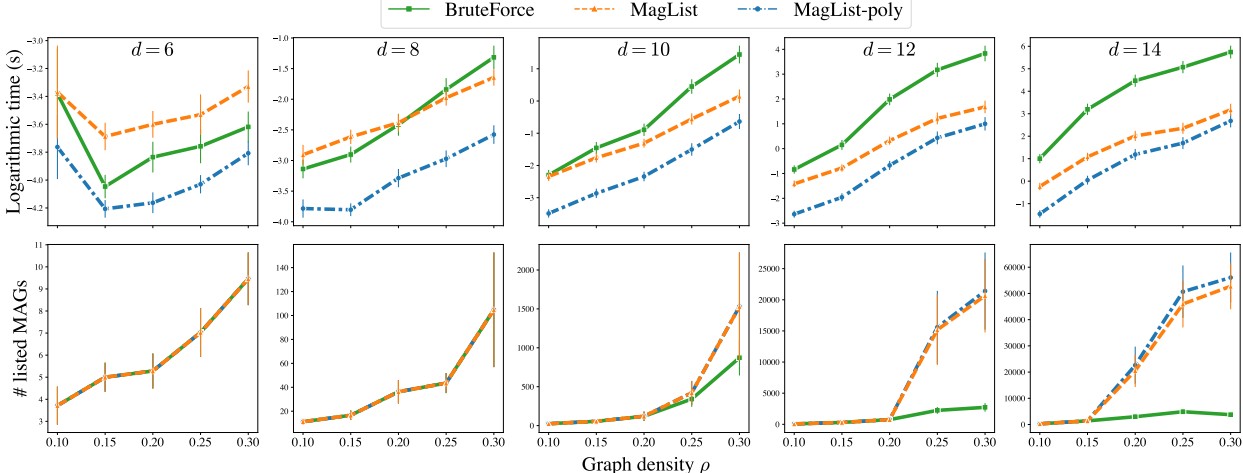

Figure 9: The logarithmic running time and number of listed MAGs within 3600 seconds for BRUTE-FORCE/MAGLIST/MAGLIST-POLY in 100 simulations under each combination of the number of vertice $d \in \{6, 8, 10, 12, 14\}$ including 3 latent variables and the graph density $\rho \in \{0.1, 0.15, 0.2, 0.25, 0.3\}$. The vertical line represents the 95% confidence interval generated by bootstrap sampling. It is determined by the 2.5th and 97.5th percentiles of 1000 estimates from the random sample of equal size with replacement from the original sample.

We then derive the PAG $\mathcal{P}$ representing the MEC to which $\mathcal{M}$ belongs and use $\mathcal{P}$ as the input for the experiment.

We implement BRUTEFORCE, MAGLIST, and MAGLIST-POLY for each PAG, recording the number of MAGs returned and the time required to list them. To handle cases with extremely long computation times, we impose a maximum running time of 3600 seconds per experiment. If the time limit is exceeded, the process is forcibly terminated, and the MAGs listed up to that point are returned.

When $d \geq 10$, the three methods may require more than 3600 seconds to complete. In such cases, the running time is capped at 3600 seconds. These results are excluded from the average runtime calculations since they do not accurately reflect actual execution times. The experimental results are shown in Fig. 9. When $d < 10$, all the methods return identical set of MAGs, demonstrating the effectiveness of MAGLIST-POLY. Moreover, MAGLIST-POLY consistently outperforms the other methods in runtime, highlighting its efficiency. These results validate that the method by transforming the circles one by one and applying the proposed rules significantly reduces computational complexity.

Finally, we conduct an experiment using real data processed from Wang et al. (2017) consisting of 7466 measurements of the abundance of phosphoproteins and phospholipids recorded under various experimental conditions. The dataset includes both observational and interventional data. We focus on causal discovery and apply our MAG listing method to support interventional variable selection by maximal entropy criterion (ME) (He & Geng, 2008). ME is compared

with random strategy (RS). For RS, a variable with circles is randomly selected to intervene. For ME, interventional variable is selected based on maximizing $H_V = \sum_{i=1}^{M} \frac{l_i}{L} \log \frac{l_i}{L}$, where $l_i$ denotes the number of MAGs with $i$-th local structure of $V$ and $L$ denotes the total number of MAGs. Three variables are randomly selected as latent variables. We calculate the number of marks that are identified by 2 interventions in the two strategies. Repeat the process above for 10 times, the number of marks identified by ME is 98, which is more than 81 by RS. It demonstrates that the maximal entropy criterion is useful in causal discovery, where the MAG listing algorithm is needed to find all MAGs.

## 7. Conclusion

In this paper, we consider a specific kind of BK, singleton BK. We present the sound and locally complete orientation rules for incorporating singleton BK into a PMG compatible with local transformation. Based on the rules, we propose the first polynomial-delay MAG listing algorithm. Experiments validate the effectiveness and efficiency. Finally, we present two examples to demonstrate the incompleteness of existing rules for general BK, and further propose two rules.

While this paper focuses on MAG listing, we highlight the five novel proposed rules. Establishing sound and complete orientation rules for incorporating general BK remains an open problem for a long time. The proposed rules in this paper can serve as a part of the complete ones in the future, which could be helpful for many areas such as causal discovery, causal effect estimation, and experimental designs.

## Acknowledgements

This research was supported by NSFC(62495092), Jiangsu Science Foundation Leading-edge Technology Program (BK20232003), Jiangsu Science Foundation (BK20241201). Tian-Zuo Wang was supported by National Postdoctoral Program for Innovative Talent and Xiaomi Foundation. We thank the reviewers for their helpful comments. In particular, we greatly appreciate meta-reviewer's valuable suggestion on discussing the necessity of local complete property for developing a polynomial delay algorithm, which inspires our discussion in Sec. 3.4. We are also grateful to Lue Tao and Zhongyi Hu for helpful discussions.

## Impact Statement

This paper presents work whose goal is to advance the field of Machine Learning. There are many potential societal consequences of our work, none which we feel must be specifically highlighted here.

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

## A. Existing Orientation Rules

In this section, we show the existing orientation rules. Zhang (2008) proposed the sound and complete rules $\mathcal{R}_1 - \mathcal{R}_{10}$ for causal discovery with observational data in the presence of latent variables. $\mathcal{R}_{12}$ is proposed by Wang et al. (2024b). $\mathcal{R}_{13}$ and $\mathcal{R}_{new}$ are proposed by Venkateswaran & Perkovic (2024). Since in this paper we only focus on the orientation rules for incorporating singleton BK, in which case $\mathcal{R}_{new}$ is not utilized, we do not denote it by a rule with a number. $\mathcal{R}_{11}$ is proposed by Wang et al. (2024b) and Venkateswaran & Perkovic (2024) independently.

$\mathcal{R}_1$: If $A \ast\!\!\to B \circ\!\!-\!\!\ast R$, and $A$ and $R$ are not adjacent, then orient the triple as $A \ast\!\!\to B \to R$.

$\mathcal{R}_2$: If $A \to B \ast\!\!\to R$ or $A \ast\!\!\to B \to R$, and $A \ast\!\!-\!\!\circ R$, then orient $A \ast\!\!-\!\!\circ R$ as $A \ast\!\!\to R$.

$\mathcal{R}_3$: If $A \ast\!\!\to B \leftarrow\!\!\ast R$, $A \ast\!\!-\!\!\circ D \circ\!\!-\!\!\ast R$, $A$ and $R$ are not adjacent, and $D \ast\!\!-\!\!\circ B$, then orient $D \ast\!\!-\!\!\circ B$ as $D \ast\!\!\to B$.

$\mathcal{R}_4$: If $\langle K, \ldots, A, B, R \rangle$ is a discriminating path between $K$ and $R$ for $B$, and $B \circ\!\!-\!\!\ast R$; then if $B \in \text{Sepset}(K, R)$, orient $B \circ\!\!-\!\!\ast R$ as $B \to R$; otherwise orient the triple $\langle A, B, R \rangle$ as $A \leftrightarrow B \leftrightarrow R$.

$\mathcal{R}_5$: For every (remaining) $A \circ\!\!-\!\!\circ R$, if there is an uncovered circle path $p = \langle A, B, \cdots, D, R \rangle$ between $A$ and $R$ s.t. $A, D$ are not adjacent and $B, R$ are not adjacent, then orient $A \circ\!\!-\!\!\circ R$ and every edge on $p$ as undirected edges.

$\mathcal{R}_6$: If $A - B \circ\!\!-\!\!\ast R$ ($A$ and $R$ may or may not be adjacent), then orient $B \circ\!\!-\!\!\ast R$ as $B \,-\!\!\ast R$.

$\mathcal{R}_7$: If $A \,-\!\!\circ B \circ\!\!-\!\!\ast R$, and $A, R$ are not adjacent, then orient $B \circ\!\!-\!\!\ast R$ as $B \,-\!\!\ast R$.

$\mathcal{R}_8$: If $A \to B \to R$, and $A \circ\!\!\to R$, orient $A \circ\!\!\to R$ as $A \to R$.

$\mathcal{R}_9$: If $A \circ\!\!\to R$, and $p = \langle A, B, D, \ldots, R \rangle$ is an uncovered possible directed path from $A$ to $R$ such that $R$ and $B$ are not adjacent, then orient $A \circ\!\!\to R$ as $A \to R$.

$\mathcal{R}_{10}$: Suppose $A \circ\!\!\to R$, $B \to R \leftarrow D$, $p_1$ is an uncovered possible directed path from $A$ to $B$, and $p_2$ is an uncovered possible directed path from $A$ to $D$. Let $U$ be the vertex adjacent to $A$ on $p_1$ ($U$ could be $B$), and $W$ be the vertex adjacent to $A$ on $p_2$ ($W$ could be $D$). If $U$ and $W$ are distinct, and are not adjacent, then orient $A \circ\!\!\to R$ as $A \to R$.

$\mathcal{R}_{11}$: Suppose an edge $A \circ\!\!-\!\!\ast B$ in a PMG $H$. Let $\mathbf{S}_A = \{V \in \mathbf{V}(H) | V \ast\!\!\to A \text{ in } H\} \cup \{A\}$. If in $H$ there is an uncovered possible directed path $\langle A, B, \cdots, K \rangle$, where $K \in \text{Anc}(\mathbf{S}_A, H)$, then orient $A \circ\!\!-\!\!\ast B$ as $A \leftarrow\!\!\ast B$.

$\mathcal{R}_{12}$: Suppose an edge $A \circ\!\!-\!\!\ast B$ in a PMG $H$. Let $\mathbf{S}_A = \{V \in \mathbf{V}(H) | V \ast\!\!\to A \text{ in } H\} \cup \{A\}$. If there is an unbridged path $\langle K_1, \cdots, K_m \rangle$ relative to $\mathbf{S}_A$ in $H[-\mathbf{S}_A]$ and for every vertex $K_i \in \{K_1, \cdots, K_m\}$, there exists an uncovered possible directed path $\langle A, B, \cdots, K_i \rangle$ ($B \neq K_i$), then orient $A \circ\!\!-\!\!\ast B$ as $A \leftarrow\!\!\ast B$.

$\mathcal{R}_{13}$: Suppose $D \ast\!\!\to C \ast\!\!-\!\!\ast A \ast\!\!-\!\!\ast D$ and $C \,-\!\!\ast B \ast\!\!-\!\!\circ A$ in a PMG $H$, if $D$ is not adjacent to $B$, then orient $A \circ\!\!-\!\!\ast B$ as $A \,-\!\!\ast B$.

$\mathcal{R}_{new}$: If $\langle Q_0, Q_1, \cdots, Q_k (= A), B \rangle$ is an almost discriminating path for $A$ in a PMG $H$ and there is $A \circ\!\!-\!\!\ast B$ in $H$, then orient $A \circ\!\!-\!\!\ast B$ as $A \to B$.

## B. Some Definitions

The *circle component* of a graph $G$ is the subgraph of $G$ that only remains all the vertices and all the circle edges. A graph is *chordal* if any cycle with more than four vertices has a chord that connects two vertices.

Next, we introduce PMG compatible with local transformation in Def. 4 proposed by Wang et al. (2024a). A PMG compatible with local transformation is initially introduced to represent the PMGs, which are obtained from incorporating local BK into PAGs and using the orientation rules to soundly and completely update the PMGs.

**Definition 4.** A PMG $\mathbb{M}$ is called a PMG *compatible with local transformation* if it satisfies the four conditions:

**(Chordal)** The circle component in $\mathbb{M}$ is chordal.

**(Balance)** For any three vertices $A, B, C$ in $\mathbb{M}$, if $A \ast\!\!\to B \circ\!\!-\!\!\ast C$, then there is an edge between $A$ and $C$ with an arrowhead at $C$, namely, $A \ast\!\!\to C$. Furthermore, if the edge between $A$ and $B$ is $A \to B$, then the edge between $A$ and $C$ is either $A \to C$ or $A \circ\!\!\to C$ (i.e., it is not $A \leftrightarrow C$). And if $A \,-\!\!\circ B \circ\!\!-\!\!\ast C$, then $A$ is adjacent to $C$. Furthermore, if $A \,-\!\!\circ B \circ\!\!-\!\!\circ C$, then $A \,-\!\!\circ C$; if $A \,-\!\!\circ B \circ\!\!\to C$, then $A \to C$ or $A \circ\!\!\to C$.

**(Complete)** For each circle at $A$ on $A \circ\!\!-\!\!* B$ in $\mathbb{M}$, there exist MAGs $\mathcal{M}_1$ and $\mathcal{M}_2$ consistent with $\mathbb{M}$ with $A \leftarrow\!\!* B$ $\in \mathbf{E}(\mathcal{M}_1)$ and $A \rightarrow\!\!* B \in \mathbf{E}(\mathcal{M}_2)$.

**(Constructive)** We can always obtain a MAG consistent with $\mathbb{M}$ by transforming $-\!\!\circ$ $/\circ\!\!\rightarrow$ to $\rightarrow$ and transforming the circle component into a DAG without new unshielded colliders.

## C. The Realization Process of MAGLIST

Here, we briefly introduce the MAG listing method MAGLIST proposed by Wang et al. (2024a). All the figures and illustrations directly refer to Wang et al. (2024a). We guide the readers to Wang et al. (2024a) for details.

MAGLIST is a recursive method. In each round, they select one vertex with circles, denoted by $X$. They enumerate all the circles at $X$ simultaneously (As discussed in Sec. 3.1, this part leads to the undesired exponential delay), which is called local structures of $X$. And they present the graphical conditions to determine whether the local structure is valid, *i.e.*, whether there is a MAG consistent with the given PAG $\mathcal{P}$ and this local structure. For each valid local structure, the local structure can be seen as a local BK, thus they use the established sound and complete orientation rules to incorporate the local BK. The implementation example of MAGLIST in the format of a search tree is shown in Fig. 10. The graph in the root node denotes a PAG $\mathcal{P}$. They aim to list all the MAGs consistent with $\mathcal{P}$. In the first round, they consider the local structures of $A$. According to their result for determining the valid local structures, they can determine that there are six valid local structures of $A$, and thus obtain six PMGs on the second line according to the marks implied by the local structures of $A$. Then they update these six graphs using the proposed rules and obtain the graphs on the third line. After the local transformation and the updates with rules, the implementation in the first round completes. In the second round, they further consider the local structures of $B$, and based on the valid local structures of $B$, they update the graph with the proposed rules. There are some PMGs without any circles, which are MAGs consistent with $\mathcal{P}$. They shade these graphs with green color. For the unshaded graphs, they are updated in the third round by considering the local structures of $C$. Some branches are omitted (those unshaded but unexpanded) for brevity. The algorithm stops until there are no new unshaded leafs.

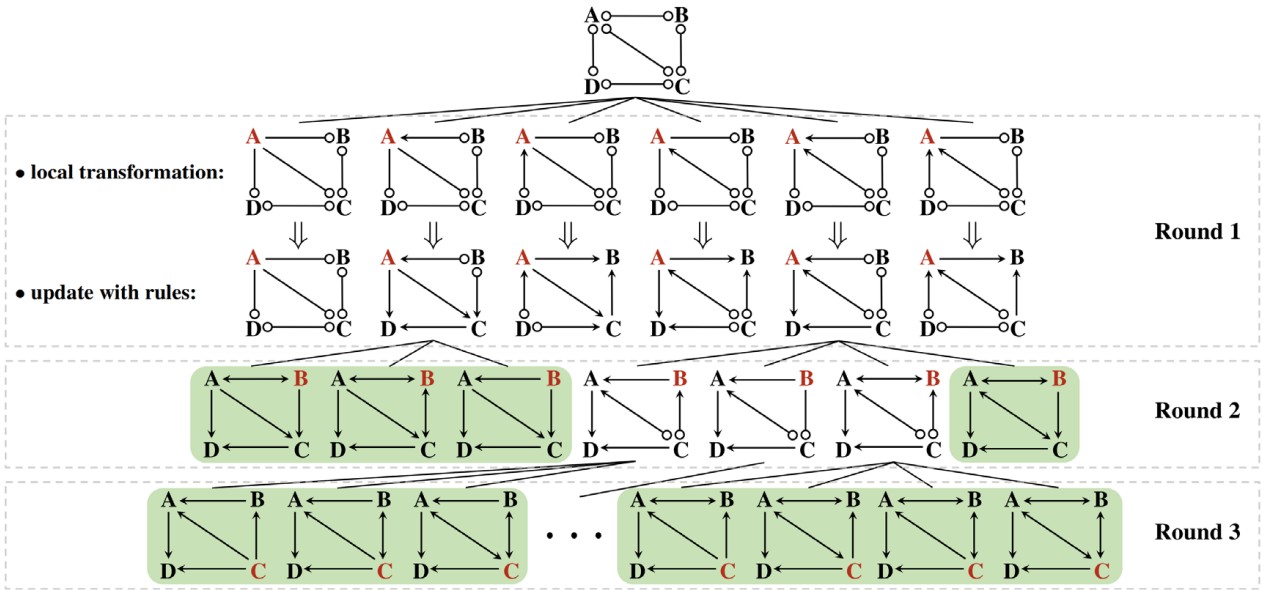

Figure 10: A realization process of MAGLIST. The graph in the root node denotes a PAG $\mathcal{P}$. The graphs in the first/second/third round are obtained from the previous round by introducing the valid local structures of $A/B/C$. There are two parts in a round: introduce the valid local structures and update the graph using the proposed orientation rules. The two parts are separately shown for only the first round. The shaded graphs denote the MAGs that are output by MAGLIST.

# D. Proofs

## D.1. Proof of Proposition 1

*Proof.* For $\mathcal{R}_{14}$, when the conditions in $\mathcal{R}_{14}$ hold, suppose $X \leftrightarrow B$ for contradiction. Following Def. 3, it is direct that if $T_1$ is prior to $B$ relative to $X$, then there exists a series of vertices $J_0(= T_1), J_1, \cdots, J_{t-1}, J_t(= B), t \geq 1$ such that for any $J_i, J_{i+1}$, one of the three conditions in Def. 3 holds. When $X \circ\!\!\ast J_t(= B)$ is oriented as $X \leftarrow\!\!\ast J_t(= B)$, if the first condition of Def. 3 holds for $J_{t-1}$ and $J_t$, there is $J_{t-1} \to J_t$, thus $X \circ\!\!\ast J_{t-1}$ can be oriented as $X \leftarrow\!\!\ast J_{t-1}$ by $\mathcal{R}_2$; if the second condition of Def. 3 holds for $J_{t-1}$ and $J_t$, $X \circ\!\!\ast J_{t-1}$ can be oriented as $X \leftarrow\!\!\ast J_{t-1}$ by $\mathcal{R}_{11}$; if the third condition of Def. 3 holds for $J_{t-1}$ and $J_t$, $X \circ\!\!\ast J_{t-1}$ can be oriented as $X \leftarrow\!\!\ast J_{t-1}$ by $\mathcal{R}_{12}$. Due to the soundness of $\mathcal{R}_2, \mathcal{R}_{11}, \mathcal{R}_{12}$, if there is $X \leftarrow\!\!\ast J_t(= B)$ and one of the three conditions in Def. 3 holds for $J_{t-1}$ and $J_t$, there must be $X \leftarrow\!\!\ast J_{t-1}$. Recursively, we can conclude that there must be $X \leftarrow\!\!\ast J_0(= T_1)$. Similarly, we can also conclude that there is $X \leftarrow\!\!\ast T_2$. In this case, if $T_1$ is not adjacent to $T_2$, there is a new unshielded collider $T_1 \ast\!\!\to X \leftarrow\!\!\ast T_2$ relative to $H$, contradiction. If there is an uncovered path $T_1 \leftarrow\!\!\circ F_1 \circ\!\!-\!\!\circ \cdots \circ\!\!-\!\!\circ F_k \circ\!\!\to T_2$ such that there is $X \circ\!\!\ast F_i$ for every $F_i, 1 \leq i \leq k$. Consider the structure comprised of $T_1, F_1, F_2, X$, where $T_1$ is not adjacent to $F_2$ and there is $T_1 \ast\!\!\to X$. According to $\mathcal{R}_{13}$, there is $F_1 \to F_2$ in any MAG consistent with $H$ and BK. Using $\mathcal{R}_1$, there is $F_1 \to F_2 \to \cdots \to F_k \to T_2$ oriented. Due to the ancestral property and $T_2 \ast\!\!\to X$, there is $F_{k-1} \ast\!\!\to X$ in $\mathcal{M}$. Hence there is an additional unshielded collider $F_{k-1} \ast\!\!\to X \leftarrow\!\!\ast T_2$ in $\mathcal{M}$ relative to $H$, which contradicts with the fact that $\mathcal{M}$ is consistent with $\mathcal{P}$. Hence $X \leftrightarrow B$ is impossible.

For $\mathcal{R}_{15}$, suppose $A \leftrightarrow R$ for contradiction. There is $A \rightarrow\!\!\ast B \to R \leftrightarrow A$, where ancestral property will always be violated no matter it is $A \to B$ or $A - B$.

$\mathcal{R}_{16}$ directly follows the ancestral property that there cannot be an arrowhead into an undirected edge. $\qquad\square$

## D.2. Proof of Theorem 1

We first introduce some existing results, which support the proof of the main theorem. $\bigoplus$ denotes concatenation of paths.

**Lemma 1** (Wang et al. (2024a))**.** *Consider a PMG $\mathbb{M}$ compatible with local transformation. If there is a possible directed path from $A$ to $B$ in $\mathbb{M}$, then there is a minimal possible directed path from $A$ to $B$ in $\mathbb{M}$.*

**Definition 5** (Bridged relative to $\mathbf{V}'$ in $H$, Wang et al. 2023b)**.** Let $H$ be a partial mixed graph. Let $G$ denote a subgraph of $H$ induced by a set of vertices $\mathbf{V}$. Given a set of vertices $\mathbf{V}'$ in $H$ that is disjoint of $\mathbf{V}$, two vertices $A$ and $B$ in the circle component of $G$ are *bridged relative to* $\mathbf{V}'$ if in each minimal circle path from $A$ to $B$ in $G$ as $V_0(= A) \circ\!\!-\!\!\circ V_1 \circ\!\!\circ \cdots \circ\!\!\circ V_n(= B)$, there exists one vertex $V_s, 0 \leq s \leq n$, such that $\mathcal{F}_i \subseteq \mathcal{F}_{i+1}, 0 \leq i \leq s-1$ and $\mathcal{F}_{i+1} \subseteq \mathcal{F}_i, s \leq i \leq n-1$, where $\mathcal{F}_i = \{V \in \mathbf{V}' \mid V \ast\!\!-\!\!\circ V_i \text{ in } H\}$. Evidently, both case $A = B$ and case that $A$ and $B$ are not connected in the circle component are the trivial cases that $A$ and $B$ in $G$ are bridged relative to $\mathbf{V}'$. Further, $G$ is *bridged relative to* $\mathbf{V}'$ *in* $H$ if any two vertices in the circle component of $G$ are bridged relative to $\mathbf{V}'$.

**Lemma 2** (Wang et al. (2024a))**.** *Suppose $\mathbb{M}$ a PMG compatible with local transformation. Given a set of vertices $\mathbf{C} \subseteq \{V | X \circ\!\!\ast V \text{ in } \mathbb{M}\}$, let $\mathbf{Z} = \{V \in \mathbf{V}(\mathbb{M}) | V = X, \text{ or there is } V \multimap \cdots \multimap V' \multimap X \text{ in } \mathbb{M} \text{ and } V' \notin \mathbf{C}\}$. There exists a MAG $\mathcal{M}$ consistent with $\mathbb{M}$ with $X \leftarrow\!\!\ast V$ for $\forall V \in \mathbf{C}$ and $X \rightarrow\!\!\ast V$ for $\forall V \in \{V \mid X \circ\!\!\ast V \text{ in } \mathbb{M}\} \backslash \mathbf{C}$ if and only if*

*(1)* $\mathrm{PossDe}(\mathbf{Z}, \mathbb{M}[-\mathbf{C}]) \cap \mathrm{Pa}(\mathbf{C}, \mathbb{M}) = \emptyset$;

*(2) the subgraph $\mathbb{M}[\mathbf{C}]$ of $\mathbb{M}$ induced by $\mathbf{C}$ is a complete graph;*

*(3)* $\mathbb{M}[\mathrm{PossDe}(\mathbf{Z}, \mathbb{M}[-\mathbf{C}]) \backslash \mathbf{Z}]$ *is bridged relative to $\mathbf{C} \cup \mathbf{Z}$ in $\mathbb{M}$;*

*(4) either $\mathbf{Z} \backslash \{X\}$ or $\{V \in \mathbf{V}(\mathbb{M}) | V \ast\!\!\to X \text{ in } \mathbb{M} \text{ or } V \in \mathbf{C}\}$ is empty.*

Next, we first present some results.

**Proposition 3.** *Denote a PMG compatible with local transformation by $\mathbb{M}$. Suppose there is a set of vertices $\mathbf{V}' \subseteq \mathbf{V}(\mathbb{M})$ and a minimal circle path $p$ in $\mathbb{M}[-\mathbf{V}']$ which is not bridged relative to $\mathbf{V}'$ as Def. 5. If no variables in $V_1, \cdots, V_m$ are ancestors of $\mathbf{V}'$ in $\mathbb{M}$, and any two vertices $C_1, C_2 \in \mathbf{V}'$ are adjacent in $\mathbb{M}$, then there must be at least one sub-path of $p$ which is an unbridged path relative to $\mathbf{V}'$ as Def. 1.*

*Proof.* According to Def. 5, suppose there is a sub-path of $p$ as $p_1 : \langle V_1, \cdots, V_m \rangle$, where $C_1 \in \mathcal{F}_1 \backslash \mathcal{F}_2$ and $C_2 \in \mathcal{F}_m \backslash \mathcal{F}_{m-1}$, where $\mathcal{F}_i = \{V \in \mathbf{V}' \mid V \ast\!\!-\!\!\circ V_i \text{ in } \mathbb{M}\}$. Without loss of generality, suppose $p_1$ is a minimal sub-path such that $p_1$ is not

bridged relative to $\mathbf{V}'$, *i.e.*, for any proper sub-path of $p_1$, it is bridged relative to $\mathbf{V}'$. Note this supposition is reasonable, since if there is a proper sub-path of $p_1$ which is not bridged relative to $\mathbf{V}'$, then we consider this path instead of $p_1$. Note for any vertex $V_i \in \{V_2, V_3, \cdots, V_m\}$, there cannot be $C_1 {*}{\rightarrow} V_i$, for otherwise there must be $C_1 {*}{\rightarrow} V, \forall V \in \{V_1, V_2, V_3, \cdots, V_m\}$ due to $\mathcal{R}_1$ and the balance property of $\mathbb{M}$, contradicting $C_1 \in \mathcal{F}_1$; there cannot be $C_1 {*}{-}{\circ} V_i$ in $\mathbb{M}$, for otherwise there is $C_1 \in \mathcal{F}_i$, such that the subpath of $p$ from $V_i$ to $V_m$ is not bridged relative to $\mathbf{V}'$, contradicting with the supposition. Hence if $V_i$ is adjacent to $C_1$, there must be $C_1 {*}{-} V_i$.

Next we consider the adjacency of each vertex and $C_1$. Suppose $V_2$ is adjacent to $C_1$. We consider the edge. Note that it has been proved to be as $C_1 {*}{-} V_2$. We first prove that the edge cannot be $C_1 - V_2$. Suppose there is $C_1 - V_2$. Due to $V_2 {\circ}{-}{\circ} V_3$, and we have proven that there is not $C_1 {*}{\rightarrow} V_3$ or $C_1 {*}{-}{\circ} V_3$ in $\mathbb{M}$, there must be $C_1 - V_3$ in $\mathbb{M}$ due to the balance and complete property. Repeatedly, we conclude that there is $C_1 - V_m$. In this case there cannot be $C_2 {*}{-}{\circ} V_m$ in $\mathbb{M}$, for otherwise there is $C_2 {*}{-} V_m$ by $\mathcal{R}_6$, contradicting with $C_2 \in \mathcal{F}_{V_m}$. Hence there cannot be $C_1 - V_2$. We then prove the impossibility of $C_1 {\circ}{-} V_2$. Consider $V_1 {\circ}{-}{\circ} V_2 {-}{\circ} C_1$ and $C_1 \in \mathcal{F}_{V_1}$, there is $V_1 {\circ}{-}{\circ} V_2 {-}{\circ} C_1 {*}{-}{\circ} V_1$, the balance property of $\mathbb{M}$ is not satisfied, contradiction. Thus $C_1 {\circ}{-} V_2$ is impossible. According to the condition, there is no $C_1 \leftarrow V_2$ since no vertex in $p$ are ancestors of $\mathbf{V}'$. Hence there cannot be $C_1 \leftarrow / {\circ}{-} / - V_2$. We conclude that $C_1$ is not adjacent to $V_2$. We then prove that $C_1$ is not adjacent to $V_3$. If they are adjacent, according to the proof above, the edge is as $C_1 {*}{-} V_3$. Note there is a sub-structure comprised of $C_1, V_1, V_2, V_3$ such that non-consecutive vertices are not adjacent except an edge between $C_1$ and $V_3$. In this case either $\mathcal{R}_5$ is triggered to transform all the edges into undirected edges, or there is an unshielded collider $V_3 {\circ}{\rightarrow} C_1 {\leftarrow}{\circ} V_1$ due to $V_1 {\circ}{-}{\circ} V_2 {\circ}{-}{\circ} V_3$. For the former case, there is $V_1 - V_2$ oriented, contradiction. For the latter case, there is $C_1 \leftarrow V_1$ oriented by $\mathcal{R}_9$, contradicting with $C_1 \in \mathcal{F}_{V_1}$. Hence $C_1$ is not adjacent to $V_3$. Recursively, we can prove that $C_1$ is not adjacent to $V_2, V_3, \cdots, V_m$. Similarly, we can prove that $C_2$ is not adjacent to $V_{m-1}, V_{m-2}, \cdots, V_1$.

Further, since $C_1$ is adjacent to $C_2$, we consider the sub-structure comprised of $C_1, V_1, V_2, \cdots, V_m, C_2$. There must be a collider in this structure, for otherwise all the edges are oriented as undirected edges by $\mathcal{R}_5$. Consider $V_1 {\circ}{-}{\circ} \cdots {\circ}{-}{\circ} V_m$, there must be a collider as $C_2 {*}{\rightarrow} C_1 {\leftarrow}{\circ} V_1$ or $C_1 {*}{\rightarrow} C_2 {\leftarrow}{\circ} V_m$. Suppose there is only $C_2 {*}{\rightarrow} C_1 {\leftarrow}{\circ} V_1$, in this case, there must be $C_2 \rightarrow C_1 \leftarrow V_1$ by $\mathcal{R}_9$, contradicting with $C_1 \in \mathcal{F}_{V_1}$. Hence there must be both $C_2 {*}{\rightarrow} C_1 {\leftarrow}{\circ} V_1$ and $C_1 {*}{\rightarrow} C_2 {\leftarrow}{\circ} V_m$. Thus there is $V_1 {\circ}{\rightarrow} C_1 \leftrightarrow C_2 {\leftarrow}{\circ} F_m$.

We conclude there is a minimal path $C_1 {\leftarrow}{\circ} V_1 {\circ}{-}{\circ} \cdots {\circ}{-}{\circ} V_m {\circ}{\rightarrow} C_2$ except for an edge $C_1 \leftrightarrow C_2$ in $\mathbb{M}$, where $C_1, C_2 \in \mathbf{V}'$, thus $p_1$ is an unbridged path relative to $\mathbf{V}'$. $\square$

**Lemma 3.** *Consider a vertex $X$ in a PMG $\mathbb{M}$ compatible with local transformation. Suppose we obtain a PMG $H$ by introducing non-empty singleton BK regarding $X$ and using the proposed orientation rules ($\mathcal{R}_1 - \mathcal{R}_{16}$). If $\mathbf{C} = \{V | V {*}{\rightarrow} X$ in $H\} \setminus \{V | V {*}{\rightarrow} X$ in $\mathbb{M}\}$, then there exists a MAG $\mathcal{M}$ consistent with $\mathbb{M}$ with $X {\leftarrow}{*} V$ for $\forall V \in \mathbf{C}$ and $X {\rightarrow}{*} V$ for $\forall V \in \{V \mid X {\circ}{-}{*} V$ in $\mathbb{M}\} \setminus \mathbf{C}$.*

*Proof.* It suffices to show that the set $\mathbf{C}$ satisfies the four conditions in Lemma 2, then we conclude the desired result according to Lemma 2.

We first prove that either $\mathbf{Z} \setminus \{X\}$ or $\{V \in \mathbf{V}(\mathbb{M}) | V {*}{\rightarrow} X$ in $\mathbb{M}$ or $V \in \mathbf{C}\}$ is empty. Suppose both $\mathbf{Z} \setminus \{X\}$ and $\{V \in \mathbf{V}(\mathbb{M}) | V {*}{\rightarrow} X$ in $\mathbb{M}$ or $V \in \mathbf{C}\}$ are not empty. There is $Z {-}{\circ} X$ and $C {*}{\rightarrow} X$ in $H$, which is impossible due to $\mathcal{R}_{16}$.

Then, we prove that the subgraph $\mathbb{M}[\mathbf{C}]$ of $\mathbb{M}$ induced by $\mathbf{C}$ is a complete graph. According to the soundness of the rules, the additional arrowheads at $X$ in $H$ relative to $\mathbb{M}$ are from either BK or the orientation rules, and the rules are sound. Hence, if $\mathbb{M}[\mathbf{C}]$ is not complete, there must be additional unshielded colliders at $X$ in any MAGs consistent with BK relative to $\mathbb{M}$. Thus there are not MAGs consistent with $\mathbb{M}$ and BK, violating the correctness of BK.

Then, we prove that $\mathrm{PossDe}(\mathbf{Z}, \mathbb{M}[-\mathbf{C}]) \cap \mathrm{Pa}(\mathbf{C}, \mathbb{M}) = \emptyset$. Suppose $F \in \mathrm{PossDe}(\mathbf{Z}, \mathbb{M}[-\mathbf{C}]) \cap \mathrm{Pa}(\mathbf{C}, \mathbb{M})$. Suppose there is $F \rightarrow C_1$ in $\mathbb{M}$, where $C_1 \in \mathbf{C}$. According to the BK, there is $C_1 {*}{\rightarrow} X$. If $F$ is adjacent to $X$ in $\mathbb{M}$, as $F \in \mathrm{PossDe}(\mathbf{Z}, \mathbb{M}[-\mathbf{C}])$, there cannot be $X {\leftarrow}{*} F$ in $H$. If there is $X {\rightarrow}{*} F$ in $H$, the structure $X {\rightarrow}{*} F \rightarrow C_1 {*}{\rightarrow} X$ violates the ancestral property. If there is $X {\circ}{-}{*} F$ in $H$, there must be $X {\circ}{-}{*} F$ in $\mathbb{M}$, $F \rightarrow C_1 {*}{\rightarrow} X {\circ}{-}{*} F$ in $\mathbb{M}$ violates the balance property of $\mathbb{M}$. Hence $F$ cannot be adjacent to $X$. However, if $F$ is not adjacent to $X$ in $\mathbb{M}$, according to Lemma 1, there is a minimal possible directed path from $X$ to $F$ in $\mathbb{M}$ and $F \rightarrow C_1 {*}{\rightarrow} X$, which is impossible since $\mathcal{R}_{11}$ would transform the circle at $X$ in the minimal possible directed path to an arrowhead. Hence that $F$ is adjacent to $X$ is also impossible. Thus there is $\mathrm{PossDe}(\mathbf{Z}, \mathbb{M}[-\mathbf{C}]) \cap \mathrm{Pa}(\mathbf{C}, \mathbb{M}) = \emptyset$.

Finally, we prove that $\mathbb{M}[\mathrm{PossDe}(\mathbf{Z}, \mathbb{M}[-\mathbf{C}]) \setminus \mathbf{Z}]$ is bridged relative to $\mathbf{C} \cup \mathbf{Z}$ in $\mathbb{M}$. Suppose $\mathbb{M}[\mathrm{PossDe}(\mathbf{Z}, \mathbb{M}[-\mathbf{C}]) \setminus \mathbf{Z}]$ is not bridged relative to $\mathbf{C} \cup \mathbf{Z}$ in $\mathbb{M}$ for contradiction. We have proven that either $\mathbf{Z} \setminus \{X\}$ or $\mathbf{C}$ is an empty set. Suppose

$\mathbf{Z} \backslash \{X\}$ is non-empty. Then there must be two vertices $Z_1, Z_2 \in \mathbf{Z}$ such that $F_1 \circ\!\!-\!\!\circ F_2 \circ\!\!-\!\!\circ \cdots \circ\!\!-\!\!\circ F_k, k \geq 2$ is not bridged relative to $\{Z_1, Z_2\}$. We omit the tedious proof for that $Z_1$ must be adjacent to $Z_2$. The main idea is, according to the definition of $\mathbf{Z}$, if there is a path $Z_1 \multimap \cdots \multimap Z_2 \multimap \cdots \multimap X$ (or swap $Z_1$ and $Z_2$ in the path), then we could prove that the sub-structure comprised of $Z_1 *\!\!-\!\!\circ F_1 \circ\!\!-\!\!\circ \cdots \circ\!\!-\!\!\circ F_k \circ\!\!-\!\!* Z_2$ and a minimal tail-circle path $Z_1 \multimap \cdots \multimap Z_2$ between $Z_1$ and $Z_2$ in $\mathbb{M}$ will never appear due to the balance and complete property of $\mathbb{M}$; if there is not such a path $Z_1 \multimap \cdots \multimap Z_2 \multimap \cdots \multimap X$, then there must be a path $Z_1 \multimap \cdots \multimap X \circ\!\!-\! \cdots \circ\!\!- Z_2$, we could prove that for each vertex between $X$ and $Z_2$ in the path, it must be adjacent to $Z_1$, for otherwise the balance and complete property of $\mathbb{M}$ is not fulfilled. Additionally, there cannot be an arrowhead into $Z_1, Z_2$, for otherwise the $\multimap$ edges will be transformed to $\rightarrow$ in $\mathbb{M}$ due to ancestral property. Hence, if there are $Z_1, Z_2 \in \mathbf{Z}$ such that $F_1 \circ\!\!-\!\!\circ F_2 \circ\!\!-\!\!\circ \cdots \circ\!\!-\!\!\circ F_k, k \geq 2$ is not bridged relative to $\{Z_1, Z_2\}$, according to Def. 5 and Prop. 3, there are $Z_1 \leftrightarrow Z_2$, which is impossible since there cannot be an arrowhead into $Z_1, Z_2$. We conclude the impossibility of $\mathbf{Z} \backslash \{X\} \neq \emptyset$.

In the following it suffices to consider the case that $\mathbf{C}$ is non-empty. Suppose there are two vertices $C_1, C_2 \in \mathbf{C}$ such that $p1 : F_1 \circ\!\!-\!\!\circ F_2 \circ\!\!-\!\!\circ \cdots \circ\!\!-\!\!\circ F_k, k \geq 2$ is not bridged relative to $\{C_1, C_2\}$. We have proved that for any two vertices $C_1, C_2 \in \mathbf{C}$, they are adjacent. Additionally, every vertex $V$ in $p_1$ cannot be an ancestor of $C_1$ in $\mathbb{M}$, for otherwise there must be $V \in \mathbf{C}$ when $C_1 \in \mathbf{C}$ according to $\mathcal{R}_2$ or $\mathcal{R}_{11}$. Similarly, we conclude $V$ is not an ancestor of $C_2$. Hence, according to Def. 5 and Prop. 3, there exists an unbridged path $p = F_1 \circ\!\!-\!\!\circ F_2 \circ\!\!-\!\!\circ \cdots \circ\!\!-\!\!\circ F_m, m \geq 2$ relative to $\{C_1, C_2\}$, where $C_1, C_2 \in \mathbf{C}$ and there is an edge $C_1 \leftrightarrow C_2$.

Due to the soundness of the orientation rules and the correctness of BK, for any MAG $\mathcal{M}$ consistent with $\mathbb{M}$ and BK, there is $C_1 *\!\!\rightarrow X \leftarrow\!\!* C_2$ in $\mathcal{M}$. Since there is $C_1 \leftarrow\!\!\circ F_1 \circ\!\!-\!\!\circ \cdots \circ\!\!-\!\!\circ F_m \circ\!\!\rightarrow C_2$ in $\mathbb{M}$, it is evident that $\forall F \in \{F_1, \cdots, F_m\}$, $F$ is an ancestor of either $C_1$ or $C_2$ in the corresponding path of $p$ in $\mathcal{M}$.

Next, we first prove that $X$ cannot be adjacent to any vertex in $F_1, \cdots, F_m$. Without loss of generality, suppose $F_i$ is adjacent to $X$. Hence, according to Prop. 3, $F_i$ is not adjacent to $C_2$ (if $i = 1$, then we consider $C_1$ instead of $C_2$). Hence there must be $C_2 *\!\!\rightarrow X \rightarrow F_i$ in $\mathcal{M}$, for otherwise there are new unshielded colliders at $X$ in $\mathcal{M}$ relative to $\mathbb{M}$. However, we have proven that $F_i$ is an ancestor of either $C_1$ or $C_2$ in the corresponding path of $p$ in $\mathcal{M}$. Given $C *\!\!\rightarrow X \rightarrow F_i \rightarrow \cdots \rightarrow C$, where $C \in \{C_1, C_2\}$, the ancestral property are not fulfilled, which contradicts with the correct BK assumption. We thus conclude that $X$ cannot be adjacent to any vertex in $F_1, \cdots, F_m$.

Next, due to $F_1 \in \text{PossDe}(\mathbf{Z}, \mathbb{M}[-\mathbf{C}]) \backslash \mathbf{Z}$, according to Lemma 1, there is a minimal possible directed path $p' = \langle X, K_1, \cdots, K_{t-1}, K_t(= F_1) \rangle, t \geq 1$ from $X$ to $F_1$ in $\mathbb{M}[\text{PossDe}(\mathbf{Z}, \mathbb{M}[-\mathbf{C}])]$. Since $p'$ is a minimal possible directed path, and $X$ is not adjacent to $F_2$, $\langle X, K_1 \rangle \bigoplus p''$ is a minimal possible directed path from $X$ to $F_2$, where $p''$ is a sub-path of $p[K_1, K_t(= F_1)] \bigoplus \langle F_1, F_2 \rangle$. Hence, there is a minimal possible directed path from $X$ to $F_2$, where $K_1$ is the vertex adjacent to $X$ in the path. Similarly, we could prove that for any vertex $F \in \{F_1, \cdots, F_m\}$, there is a minimal possible directed path from $X$ to $F$, where $K_1$ is the vertex adjacent to $X$ in the path. In this case, $\mathcal{R}_{12}$ should be triggered to orient $X \leftarrow\!\!* K_1$, that is $K_1 \in \mathbf{C}$, contradicting with $K_1 \in \text{PossDe}(\mathbf{Z}, \mathbb{M}[-\mathbf{C}])$.

Hence, the four conditions in Lemma 2 are satisfied. We conclude the desired result according to Lemma 2. □

**Lemma 4.** *Consider a vertex $X$ in a PMG $\mathbb{M}$ compatible with local transformation. Suppose we obtain a PMG $H$ by introducing non-empty singleton BK regarding $X$ and using the proposed orientation rules ($\mathcal{R}_1 - \mathcal{R}_{16}$). If there is an edge $X \circ\!\!-\!\!* T$ in $H$, then there is a MAG $\mathcal{M}$ consistent with $\mathbb{M}$ and BK such that there is $X \leftarrow\!\!* T$ in $\mathcal{M}$.*

*Proof.* According to Lemma 3, we could always obtain a MAG $\mathcal{M}$ with $X \leftarrow\!\!* V$ for $\forall V \in \mathbf{C}$ and $X \rightarrow\!\!* V$ for $\forall V \in \{V \mid X \circ\!\!-\!\!* V$ in $\mathbb{M}\} \backslash \mathbf{C}$ by the procedure in the proof of Lemma 2. Consider the edge $X \circ\!\!-\!\!* T$ in $H$. We will prove that there is a MAG $\mathcal{M}$ with an edge $X \leftarrow\!\!* T$ consistent with $\mathbb{M}$ and the BK. Suppose we can obtain a PMG $H'$ from $H$ by introducing $X \leftarrow\!\!* T$ with the orientation rules $\mathcal{R}_2, \mathcal{R}_{11}, \mathcal{R}_{12}, \mathcal{R}_{16}$. Note here when we apply $\mathcal{R}_{16}$, we only consider the case that if there is $A *\!\!\rightarrow B \circ\!\!- R$, then orient $B \circ\!\!- R$ as $B \leftarrow R$, as here we only introduce an additional arrowhead at $X$ and care about the identification of the marks at $X$.

Denote $\mathbf{C}' = \{V \mid V *\!\!\rightarrow X$ in $H'\} \backslash \{V \mid V *\!\!\rightarrow X$ in $\mathbb{M}\}$. Evidently, we can divide $\mathbf{C}'$ into two disjoint sets $\mathbf{C}_1$ and $\mathbf{C}_2$, where $\mathbf{C}_1$ denotes the set of vertices $\mathbf{C}$ in Lemma 3, and $\mathbf{C}_2$ denotes the set of vertices $\{V \mid V *\!\!\rightarrow X$ in $H'\} \backslash \{V \mid V *\!\!\rightarrow X$ in $H\}$. As we introduce $X \leftarrow\!\!* T$ based on $H$, $\mathbf{C}_2$ is not empty. It suffices to show that there is a MAG $\mathcal{M}$ consistent with $\mathbb{M}$ such that $\mathbf{C}' = \{V \mid V *\!\!\rightarrow X$ in $\mathcal{M}\} \backslash \{V \mid V *\!\!\rightarrow X$ in $\mathbb{M}\}$. Next, we adopt the similar ideas with Lemma 3. We will prove that $\mathbf{C}'$ fulfills the four conditions in Lemma 2. Note that the proof here is quite different from that of Lemma 3, as here there is no correct BK assumption when we further incorporate $X \leftarrow\!\!* T$. Note according to the definition of $\mathbf{Z}$ that

$\mathbf{Z} = \{V \in \mathbf{V}(\mathbb{M}) | V = X$, or there is $V \multimap \cdots \multimap V' \multimap X$ in $\mathbb{M}$ and $V' \notin \mathbf{C}\}$, $\mathbf{Z}$ relies on $\mathbf{C}$. When we consider $\mathbf{C}'$, to distinguish them, we denote $\mathbf{Z}' = \{V \in \mathbf{V}(\mathbb{M}) | V = X$, or there is $V \multimap \cdots \multimap V' \multimap X$ in $\mathbb{M}$ and $V' \notin \mathbf{C}'\}$.

We first prove that either $\mathbf{Z}' \backslash \{X\}$ or $\{V \in \mathbf{V} | V \ast\!\!\rightarrow X$ in $\mathbb{M}$ or $V \in \mathbf{C}'\}$ is empty. Suppose both $\mathbf{Z}' \backslash \{X\}$ and $\{V \in \mathbf{V} | V \ast\!\!\rightarrow X$ in $\mathbb{M}$ or $V \in \mathbf{C}'\}$ are not empty. Suppose in $H$ there is $Z \multimap X$. According to Lemma 3, there is not a vertex $C \in \mathbf{C}_1$ such that there is $C \ast\!\!\rightarrow X$ in $H$. It suffices to consider $C \in \mathbf{C}_2$. In this case there must be $X \circ\!\!-\!\ast C$ in $H$. $C$ must be adjacent to $Z$, for otherwise there is $X \rightarrow\!\!\ast C$ in $H$ due to $\mathcal{R}_7$, contradiction. In this case, there is $Z \rightarrow X$ in $H'$ due to $\mathcal{R}_{16}$, thus $Z \in \mathbf{C}_2$. There cannot be $Z \in \mathbf{Z}'$, contradiction. Hence, for any vertex $V$ with an edge $V \multimap X$ in $H$, there is $V \rightarrow X$ in $H'$. In this case $\mathbf{Z}' \backslash \{X\} = \emptyset$, contradiction. We get the desired result.

Then, we will prove that $\mathrm{PossDe}(\mathbf{Z}', \mathbb{M}[-\mathbf{C}']) \cap \mathrm{Pa}(\mathbf{C}', \mathbb{M}) = \emptyset$. Suppose $F \in \mathrm{PossDe}(\mathbf{Z}', \mathbb{M}[-\mathbf{C}']) \cap \mathrm{Pa}(\mathbf{C}', \mathbb{M})$. Evidently $\mathbf{C}'$ is not empty, thus $\mathbf{Z}' \backslash \{X\}$ is empty and $F \in \mathrm{PossDe}(X, \mathbb{M}[-\mathbf{C}']) \cap \mathrm{Pa}(\mathbf{C}', \mathbb{M})$. Suppose there is $C \in \mathbf{C}'$ such that there is $F \rightarrow C$. According to the proof of Lemma 3, $C \notin \mathbf{C}_1$, thus there is $C \in \mathbf{C}_2$. Due to $F \in \mathrm{PossDe}(X, \mathbb{M}[-\mathbf{C}'])$, there is a minimal possible directed path $p$ from $X$ to $F$ in $\mathbb{M}$. Suppose $p = \langle X, F_1, \cdots, F_k(= F)\rangle, k \geq 1$. As $F \in \mathrm{PossDe}(X, \mathbb{M}[-\mathbf{C}'])$ and $\mathbb{M}$ satisfies the complete property, $p$ is either $X \rightarrow\!\!\ast F_1 \cdots \rightarrow\!\!\ast F_k$ in $\mathbb{M}$ or $X \circ\!\!-\!\ast F_1 \cdots F_k$ in $\mathbb{M}$. For the former case, there cannot be $X \circ\!\!-\!\ast C$ in $\mathbb{M}$, for otherwise the complete property of $\mathbb{M}$ is violated since there cannot be $X \leftarrow\!\!\ast C$ in any MAGs consistent with $\mathbb{M}$ given $F_k \rightarrow C$. Thus there must be $X \rightarrow C$ in $\mathbb{M}$, which contradicts with $C \in \mathbf{C}_2$. For the latter case, since $C \in \mathbf{C}_2$, if there is $X \circ\!\!-\!\ast F_1$ in $H$, $X \circ\!\!-\!\ast F_1$ should be oriented as $X \leftarrow\!\!\ast F_1$ by $\mathcal{R}_2$ or $\mathcal{R}_{11}$, which implies that $F_1 \in \mathbf{C}_2$, contradicting with $F_1 \in \mathrm{PossDe}(\mathbf{Z}', \mathbb{M}[-\mathbf{C}'])$. If there is $X \rightarrow\!\!\ast F_1$ in $H$, there must be $F_{k-1} \rightarrow C, F_{k-2} \rightarrow C, \cdots, X \rightarrow C$ in $H$ triggered by $\mathcal{R}_{15}$ due to $F_k \rightarrow C$ (we omit the details), contradicting with $C \in \mathbf{C}_2$. Hence $F \in \mathrm{PossDe}(\mathbf{Z}', \mathbb{M}[-\mathbf{C}']) \cap \mathrm{Pa}(\mathbf{C}', \mathbb{M})$ is impossible.

Then, we prove that the subgraph $\mathbb{M}[\mathbf{C}']$ is a complete graph. According to the proof of Lemma 3, for any vertex $C_1, C_2 \in \mathbf{C}_1$, $C_1$ are adjacent to $C_2$; for any vertex $C_1 \in \mathbf{C}_1$ and $C_2 \in \mathbf{C}_2$, $C_1$ are adjacent to $C_2$, for otherwise there is $X \rightarrow C_2$ in $H$ due to $C_1 \ast\!\!\rightarrow X$, contradicting with $C_2 \in \mathbf{C}_2$ which means that there is $X \circ\!\!-\!\ast C_2$ in $H$.

We then prove that for $C_1, C_2 \in \mathbf{C}_2$, it is impossible that $C_1$ is not adjacent to $C_2$. Note $X \circ\!\!-\!\ast T$ is firstly oriented as $X \leftarrow\!\!\ast T$ based on $H$. Since we only use the orientation rules $\mathcal{R}_2, \mathcal{R}_{11}, \mathcal{R}_{12}, \mathcal{R}_{16}$ to obtain $H'$ based on $H$, the other additional arrowheads at $X$ are only triggered by $\mathcal{R}_2, \mathcal{R}_{11}, \mathcal{R}_{12}, \mathcal{R}_{16}$. We first consider the case that $C_1, C_2 \neq T$. We consider which rules triggers the transformation of $X \circ\!\!-\!\ast C_1$ to $X \leftarrow\!\!\ast C_1$. For simplicity, we say vertex $C$ is triggered by rule $\mathcal{R}_i$ if $\mathcal{R}_i$ triggers the transformation of $X \circ\!\!-\!\ast C$ to $X \leftarrow\!\!\ast C$. It is impossible that there is $C_1 \multimap X \circ\!\!-\! C_2$ in $H$, for otherwise there is $C_1 - X - C_2$ in $H$ due to $\mathcal{R}_7$. Note after we orient $X \circ\!\!-\!\ast T$ as $X \leftarrow\!\!\ast T$, we only apply $\mathcal{R}_2, \mathcal{R}_{11}, \mathcal{R}_{12}, \mathcal{R}_{16}$, in which process there are no tails introduced. Hence if there is $C_1 \multimap X$ in $H'$, there must be $C_1 \multimap X$ in $H$. Similarly, if there is $C_2 \multimap X$ in $H'$, there must be $C_2 \multimap X$ in $H$. Thus there is $C_1 \multimap X \circ\!\!-\! C_2$ in $H$, contradiction. Hence it is impossible that both $C_1, C_2$ are triggered by $\mathcal{R}_{16}$.

If $C_1$ is triggered by $\mathcal{R}_{16}$ and $C_2$ is triggered by $\mathcal{R}_2, \mathcal{R}_{11}, \mathcal{R}_{12}$. As we do not introduce new tails in $H'$ relative to $H$, there is $C_1 \multimap X$. Since $C_1$ is not adjacent to $C_2$, hence there cannot be $C_2 \ast\!\!-\!\!\circ X$, contradiction. Hence neither of $C_1, C_2$ could be triggered by $\mathcal{R}_{16}$.

Note if a vertex $V$ is triggered by $\mathcal{R}_{16}$, then an edge $X \circ\!\!-\! V$ is oriented as $X \leftarrow V$. Note this edge will not be a necessary part of the sub-structures where $\mathcal{R}_2, \mathcal{R}_{11}, \mathcal{R}_{12}$ are triggered, because the necessary arrowhead at $X$ that activates $\mathcal{R}_2, \mathcal{R}_{11}, \mathcal{R}_{12}$ is on an edge in the form of $X \circ\!\!\rightarrow V$. Hence, in the following, we only need to consider the transformation triggered by $\mathcal{R}_2, \mathcal{R}_{11}, \mathcal{R}_{12}$. Then, according to Def. 3, if $X \leftarrow\!\!\ast T$ triggers orienting $X \circ\!\!-\!\ast T'$ as $X \leftarrow\!\!\ast T'$ by $\mathcal{R}_2, \mathcal{R}_{11}$, or $\mathcal{R}_{12}$, then $T'$ is prior to $T$. Recursively, we conclude that either (a) $C_1$ and $C_2$ are prior to $T$ or (b) $C_1$ or $C_2$ is $T$. If both $C_1$ and $C_2$ are prior to $T$, $C_1$ and $C_2$ are not adjacent, then according to $\mathcal{R}_{14}$, there is $X \rightarrow T$ in $H$, contradicting with $X \circ\!\!-\!\ast T$ in $H$.

Then we consider the case that one of $C_1, C_2$ is $T$. Without loss of generality, suppose $C_1 = T$. $\mathcal{R}_2, \mathcal{R}_{11}$, and $\mathcal{R}_{12}$ cannot trigger a transformation of $X \circ\!\!-\!\ast C_2$ to $X \leftarrow\!\!\ast C_2$ due to $X \leftarrow\!\!\ast C_1$ where $C_1$ and $C_2$ are not adjacent. If $C_2$ is triggered by $\mathcal{R}_{16}$, then there is $X \circ\!\!-\! C_2$ in $H$, in which case there cannot be $X \circ\!\!-\! C_1(= T)$ in $H$ and $C_1$ is not adjacent to $C_2$, contradiction.

Till now, we have proven the impossibility of all possible cases that $\mathbf{C}_1$ is triggered by $\mathcal{R}_2, \mathcal{R}_{11}, \mathcal{R}_{12}, \mathcal{R}_{16}$ and $\mathbf{C}_2$ is triggered by $\mathcal{R}_2, \mathcal{R}_{11}, \mathcal{R}_{12}, \mathcal{R}_{16}$. Hence $\mathbb{M}[\mathbf{C}']$ is a complete graph.

In the following we will prove that $\mathbb{M}[\mathrm{PossDe}(\mathbf{Z}', \mathbb{M}[-\mathbf{C}']) \backslash \mathbf{Z}']$ is bridged relative to $\mathbf{C}' \cup \mathbf{Z}'$ in $\mathbb{M}$. Suppose $\mathbb{M}[\mathrm{PossDe}(\mathbf{Z}', \mathbb{M}[-\mathbf{C}']) \backslash \mathbf{Z}']$ is not bridged relative to $\mathbf{C}' \cup \mathbf{Z}'$ in $\mathbb{M}$ for contradiction. Based on the similar proof process as that for Lemma 3, according to Def. 5 and Prop. 3, there must exist an unbridged path $p = F_1 \circ\!\!-\!\!\circ F_2 \circ\!\!-\!\!\circ \cdots \circ\!\!-\!\!\circ F_m, m \geq 2$

relative to $\{C_1, C_2\}$, where $C_1, C_2 \in \mathbf{C}'$.

We then prove that $X$ cannot be adjacent to any vertex in $F_1, \cdots, F_m$. Suppose $X$ is not adjacent to $F_1, \cdots, F_{i-1}$ but adjacent to $F_i, i \geq 2$. There is a sub-structure comprised $X, C_1, F_1, \cdots, F_i, X$, where any two non-consecutive vertices are not adjacent. Note there must be a collider in this sub-structure and $F_1 \circ\!\!-\!\!\circ F_2 \circ\!\!-\!\!\circ \cdots \circ\!\!-\!\!\circ F_i$ in $\mathbb{M}$. If there is not collider $C_1 *\!\!\rightarrow X \leftarrow\!\!* F_i$ in $\mathbb{M}$, there will be only a collider $F_1 \rightarrow C_1 \leftarrow F_i$ due to $\mathcal{R}_9$, contradicting with $C_1 \in \mathbf{C}'$. Hence there is $X \leftarrow\!\!* F_i$ in $\mathbb{M}$, contradicting with $F_i \in \text{PossDe}(\mathbf{Z}', \mathbb{M}[-\mathbf{C}']) \backslash \mathbf{Z}'$. Hence if $X$ is adjacent to some vertex in $F_1, \cdots, F_m$, then $X$ is adjacent to any vertex in $F_1, \cdots, F_m$.

Here we consider the case that $X$ is adjacent to $F_1, \cdots, F_m$. According to the proof of Lemma 3, we have proven the impossibility of $C_1, C_2 \in \mathbf{C}_1$. If $C_1 \in \mathbf{C}_1$ and $C_2 \in \mathbf{C}_2$, considering $C_1$ is not adjacent to $F_2$, there must be $X \rightarrow F_2$ and $F_1 \rightarrow F_2 \cdots \rightarrow \cdots F_m \rightarrow C_2$ in $H$ according to $\mathcal{R}_{13}$. Thus there is $X \rightarrow C_2$ according to $\mathcal{R}_8$, contradicting with $X \circ\!\!-\!\!* C_2$ in $H$. We then consider the case that $C_1, C_2 \in \mathbf{C}_2$. Due to the complete graph $\mathbb{M}[\mathbf{C}']$, $C_1$ is adjacent to $C_2$. We consider the sub-structure comprised of $C_1, F_1, \cdots, F_m, C_2$, where any two non-consecutive vertices are not adjacent except for an edge connecting $C_1$ and $C_2$. In this case, if there is no unshielded collider $F_1 *\!\!\rightarrow C_1 \leftarrow\!\!* C_2$, there must be $F_m \rightarrow C_2 \leftarrow C_1$ in $\mathbb{M}$ by $\mathcal{R}_9$, which contradicts with $C_2 \in \mathcal{F}_m$. Hence, there is an unshielded collier $F_1 *\!\!\rightarrow C_1 \leftarrow\!\!* C_2$ in $\mathbb{M}$. Similarly, there is an unshielded collider $F_m *\!\!\rightarrow C_2 \leftarrow\!\!* C_1$ in $\mathbb{M}$. Hence there is $C_1 \leftrightarrow C_2$ in $\mathbb{M}$. In the following, we will prove the impossibility of $C_1 \leftrightarrow C_2$ in $\mathbb{M}$.

We first consider the case $C_1 = T$ or $C_2 = T$. Without loss of generality, suppose $C_1 = T$. Consider the transformation series is as $J_0(= T), J_1, \cdots, J_t(= C_2), t \geq 2$, i.e., when orienting $X \circ\!\!-\!\!* T$ as $X \leftarrow\!\!* T$, the orientation of $X \circ\!\!-\!\!* J_{i-1}$ triggers the transformation of $X \circ\!\!-\!\!* J_i$ to $X \leftarrow\!\!* J_i$ by $\mathcal{R}_2, \mathcal{R}_{11}, \mathcal{R}_{12}$. Next, we will prove the impossibility of $C_1 \leftrightarrow C_2$. We first consider the edge connecting $J_0$ and $J_1$. When $X \circ\!\!-\!\!* J_0$ is oriented as $X \leftarrow\!\!* J_0$, $X \circ\!\!-\!\!* J_1$ is oriented as $X \leftarrow\!\!* J_1$ by $\mathcal{R}_2, \mathcal{R}_{11}, \mathcal{R}_{12}$ (there are arrowheads at $C_2$ thus $\mathcal{R}_{16}$ cannot trigger the transformation, and we have proven before that $\mathcal{R}_{16}$ will not trigger the transformation of $\mathcal{R}_2, \mathcal{R}_{11}, \mathcal{R}_{12}$). We discuss the rules that trigger the orientation. If it is $\mathcal{R}_2$, then there must be $J_1 \rightarrow J_0$ in $\mathbb{M}$. If it is $\mathcal{R}_{11}$, then there must $J_1 \circ\!\!\rightarrow J_0$ in $\mathbb{M}$ (we omit some details here, which can be proved according to the balance and completer property of $\mathbb{M}$). If it is $\mathcal{R}_{12}$, then there must be $J_1 \circ\!\!\rightarrow J_0$ in $\mathbb{M}$. Hence, there is $J_1 \rightarrow J_0$ or $J_1 \circ\!\!\rightarrow J_0$ in $\mathbb{M}$. Similarly, we could prove that there is $J_2 \rightarrow J_1$ or $J_2 \circ\!\!\rightarrow J_1$. As $\mathbb{M}[\mathbf{C}']$ is a complete graph, $J_0$ must be adjacent to $J_2$. According to the balance property and the rules, there can only be $J_2 \rightarrow J_0$ or $J_2 \circ\!\!\rightarrow J_0$ in $\mathbb{M}$. Recursively, for any vertex $J_i, 1 \leq i \leq t$, we could prove that there is $J_i \rightarrow J_0$ or $J_i \circ\!\!\rightarrow J_0$ in $\mathbb{M}$. Hence there is $J_t(= C_2) \rightarrow J_0(= T)$ or $J_t(= C_2) \circ\!\!\rightarrow J_0(= T)$, which contradicts with $C_2 \leftrightarrow T$ in $\mathbb{M}$. Hence $C_1 = T$ or $C_2 = T$ is impossible.

If $C_1 \neq T$ and $C_2 \neq T$, then there is an unbridged path $\langle F_1, \cdots, F_k \rangle$ relative to $\{C_1, C_2\}$ such that $X \circ\!\!-\!\!* F_i$ for every $F_i, 1 \leq i \leq m$. According to $\mathcal{R}_{14}$, there should be $X \rightarrow T$ in $H$, contradicts with $X \circ\!\!-\!\!* T$ in $H$. Hence, $X$ cannot be adjacent to any vertex in $F_1, \cdots, F_m$.

Next, due to $F_1 \in \text{PossDe}(\mathbf{Z}', \mathbb{M}[-\mathbf{C}'])$, according to Lemma 1, there is a minimal possible directed path $p' = \langle X, K_1, \cdots, K_{t-1}, K_t(= F_1) \rangle, t \geq 1$ from $X$ to $F_1$ in $\mathbb{M}[\text{PossDe}(\mathbf{Z}', \mathbb{M}[-\mathbf{C}'])]$. Since $p'$ is a minimal possible directed path, and $X$ is not adjacent to $F_2$, $\langle X, K_1 \rangle \bigoplus p''$ is a minimal possible directed path from $X$ to $F_2$, where $p''$ is a sub-path of $p[K_1, K_t(= F_1)] \bigoplus \langle F_1, F_2 \rangle$. Hence, there is a minimal possible directed path from $X$ to $F_2$, where $K_1$ is the vertex adjacent to $X$ in the path. Similarly, we could prove that for any vertex $F \in \{F_1, \cdots, F_m\}$, there is a minimal possible directed path from $X$ to $F$, where $K_1$ is the vertex adjacent to $X$ in the path. In this case, $\mathcal{R}_{12}$ should be triggered to orient $X \leftarrow\!\!* K_1$, that is $K_1 \in \mathbf{C}'$, contradicting with $K_1 \in \text{PossDe}(\mathbf{Z}', \mathbb{M}[-\mathbf{C}'])$.

Hence, we have proven that the four conditions in Lemma 2 are satisfied. We conclude the desired result according to Lemma 2. There is a MAG $\mathcal{M}$ consistent with $\mathbb{M}$ and BK such that there is $X \leftarrow\!\!* T$ in $\mathcal{M}$. $\quad\square$

*Proof of Theorem 1.* Combining Lemma 3 and Lemma 4, we conclude the desired result in Theorem 1. $\quad\square$

### D.3. Proof of Theorem 2

*Proof.* For MAGLIST-POLY and MAGLIST, they have the same outer loop that selects a vertex with circles and transforms all the circles at this vertex. The only difference is, for MAGLIST, in each outer loop they enumerate all possible local structures of the selected vertex and determine the validity of each local structure, while for MAGLIST-POLY, there is an inner loop as shown in Function LOCALTRANSFORM of Alg. 1 to find all valid local structures. We will get the desired result by proving that MAGLIST-POLY and MAGLIST obtain the same set of PMGs after each round of outer loop. Suppose

in this outer loop, the input graph is a PMG $\mathbb{M}$ compatible with local transformation. Without loss of generality, suppose in this round, we select vertex $X$ and transform the circles at $X$. We consider MAGLIST-POLY in the following. We will prove (a) by the inner loop MAGLIST-POLY can find all the valid local structures of $X$ in the form of PMGs on Line 17 of Alg. 1, where a valid local structure of $X$ implies that there is a MAG $\mathcal{M}$ consistent with $\mathbb{M}$ such that $\mathcal{M}$ has this local structure of $X$, (b) the obtained graphs on Line 17 of Alg. 1 are PMGs compatible with local transformation.

(a) Denote $\mathbf{C} \subseteq \{V | V \ast\!\!-\!\!\circ X \text{ in } \mathbb{M}\}$. For a given set $\mathbf{C}$, it dictates a local structure of $X$ based on $\mathbb{M}$ such that the circle at $X$ on an edge between $X$ and $V$ is oriented as an arrowhead if $V \in \mathbf{C}$ and the circle is oriented as a tail if $V \notin \mathbf{C}$. Hence, we use $\{\mathbf{C}\}_{ours}$ and $\{\mathbf{C}\}_{local}$ to represent the local structures obtained by LOCALTRANSFORM and Wang et al. (2024a), respectively. In the following we will prove that $\{\mathbf{C}\}_{ours} \overset{set}{=} \{\mathbf{C}\}_{local}$.

Since $\mathbb{M}$ is a PMG compatible with local transformation, $\mathbb{M}$ fulfills the complete condition, *i.e.*, for every circle at $X$, it could be an arrowhead in a MAG $\mathcal{M}_1$ consistent with $\mathbb{M}$ and a tail in a MAG $\mathcal{M}_2$ consistent with $\mathbb{M}$. Hence it is valid to enumerate it as an arrowhead and a tail, and the proposed rules are sound to incorporate singleton BK into $\mathbb{M}$ according to Prop. 1. Without loss of generality, suppose we first enumerate $X \circ\!\!-\!\!\ast V_1$ as $X \leftarrow\!\!\ast V_1$ and obtain an updated graph $H_1$ using the proposed orientation rules. If there are circles at $X$ in $H_1$, suppose it is $X \circ\!\!-\!\!\ast V_2$ without loss of generality, as the proposed rules are locally complete for incorporating singleton BK into a PMG compatible with local transformation, there must be MAGs consistent with $H_1$ with $X \leftarrow\!\!\ast V_2$ and $X \rightarrow\!\!\ast V_2$. Hence it is valid to enumerate the circle at $X$ on the edge between $X$ and $V_2$ as an arrowhead and a tail, and the proposed rules are also sound to incorporate singleton BK into $\mathbb{M}$. Suppose we enumerate $X \circ\!\!-\!\!\ast V_2$ as $X \leftarrow\!\!\ast V_2$ and obtain an updated graph $H_2$ using the proposed orientation rules. Essentially, the process above is that we incorporate the singleton BK $V_1 \ast\!\!\rightarrow X \leftarrow\!\!\ast V_2$ into $\mathbb{M}$. Due to the locally completeness of the proposed rules, it is also valid to enumerate another circle at $X$ as an arrowhead and a tail. Repeat the process above, for all the obtained local structures, it must be a valid local structure since the rules are locally complete, thus $\{\mathbf{C}\}_{ours} \subseteq \{\mathbf{C}\}_{local}$.

For a valid local structure represented by $\mathbf{C}$ in $\{\mathbf{C}\}_{local}$, without loss of generality, suppose $\mathbf{C} = \{V_1, V_2, \cdots, V_k\}$. Consider we enumerate the circles at $X$. In the first round, when we enumerate $X \circ\!\!-\!\!\ast V_1$ as $X \leftarrow\!\!\ast V_1$, as the rules are sound, we will not introduce a mark which contradicts with the valid local structure dictated by $\mathbf{C}$. Hence, there is either $X \circ\!\!-\!\!\ast V_2$ or $X \leftarrow\!\!\ast V_2$ obtained by using the orientation rules. For the former case, we continue considering the circle at $X$ on the edge between $X$ and $V_2$. By enumerating it as an arrowhead and a tail, we will obtain a local structure with $X \leftarrow\!\!\ast V_2$. And since the rules are sound and locally complete, we will not introduce a mark which contradicts with the valid local structures dictated by $\mathbf{C}$. For the later case, there have been $X \leftarrow\!\!\ast V_2$. Repeat the process above for $V_1, V_2, \cdots, V_d$, LOCALTRANSFORM can find a local structure represented by $\mathbf{C}$. Thus there is $\{\mathbf{C}\}_{ours} \supseteq \{\mathbf{C}\}_{local}$.

Hence there is $\{\mathbf{C}\}_{ours} \overset{set}{=} \{\mathbf{C}\}_{local}$. That implies that LOCALTRANSFORM can find all and only the valid local structures of $X$.

(b) In the process of updating the graph based on singleton BK regarding $X$ with the proposed orientation rules, we use the rules $\mathcal{R}_1 - \mathcal{R}_{16}$. And after transforming all the circles at $X$, essentially, we introduce the local BK regarding $X$. Note the rules used in MAGLIST are a proper subset of $\mathcal{R}_1 - \mathcal{R}_{16}$, and Wang et al. (2024a) proved that the rules are sound and complete for incorporating local BK into a PMG compatible with local transformation (Theorem 2 of Wang et al. (2024a)). Hence, by using $\mathcal{R}_1 - \mathcal{R}_{16}$ or the rules of Wang et al. (2024a), we can obtain the same PMGs. Since the obtained graph by MAGLIST is a PMG compatible with local transformation, the obtained graph on Line 17 of Alg. 1 is also a PMG compatible with local transformation.

In light of the fact that the input PAG $\mathcal{P}$ is a PMG compatible with local transformation, given the results (a) and (b) above, we get the desired result by mathematical induction. $\qquad\square$

### D.4. Proof of Theorem 3

*Proof.* According to Alg. 1, there are two nested loops. In each outer loop, we select one variable/vertex with circles. Then, in each inner loop, we transform a circle at this vertex. Without loss of generality, suppose in the $i$-th round of outer loop, the circles at $V_i$ are transformed to non-circles. And we suppose the outer loop runs for more than one round. It is a trivial case that the outer loop finishes in one round.

As discussed in the main paper, the complexity of implementing the orientation rules is $\mathcal{O}(m^3 d^4)$, where $m$ denotes the number of edges and $d$ denotes the number or vertices. For brevity, we denote this complexity by $S$.

We first consider the complexity for an inner loop, *i.e.*, the complexity of implementing Line 9 of Alg. 1 in one round. In this round $V_1$ is selected. Suppose there are $k_1$ kinds of local structures of $V_1$. Hence, by implementing Line 9 of Alg. 1, we will find $k_1$ PMGs. When the function LOCALTRANSFORM is triggered, a circle at $V_1$ will be enumerated as an arrowhead and a tail, hence we could use a binary tree $\mathcal{T}_{within}$ to denote the process of finding all $k_1$ PMGs, where each node denotes a PMG obtained by enumerating a circle at $V_1$ and using the orientation rules, and only the leaf node denotes the PMGs with no-circles at $V_1$. The binary tree is as Fig. 6, with only the difference that in the binary tree there are only the second and third graphs in the second layer. Since the proposed rules are locally complete, every non-leaf node must have two children nodes, *i.e.*, there cannot be a non-leaf node with only one child node. In $\mathcal{T}_{inner}$, denote the number of edges and number of nodes by $\|\mathbf{E}(\mathcal{T}_{inner})\|$ and $\|\mathbf{V}(\mathcal{T}_{inner})\|$. Denote the number of leaf nodes and the nodes with two children by $n_0$ and $n_2$. There is $\|\mathbf{E}(\mathcal{T}_{inner})\| = \|\mathbf{V}(\mathcal{T}_{inner})\| - 1 = n_0 + n_2 - 1$ and $\|\mathbf{E}(\mathcal{T}_{inner})\| = 2 * n_2$. Hence, $n_2 = n_0 - 1$. Since $n_0 = k_1$, there is $\|\mathbf{E}(\mathcal{T}_{inner})\| = 2k_1 - 2$. In the process of obtaining each PMG from the PMG in the parent node, the proposed rules are incorporated. Hence the total complexity is $(2k_1 - 2)S$.

We then consider the total complexity of implementing Alg. 1. After we incorporate the singleton BK regarding $V_1$, we will transform the circles at $V_2, \cdots, V_d$. Note that there are $k_1$ local structures of $X$ since there are $k_1$ PMGs obtained in the inner loop, where a local structure implies an orientation of all the circles at $X$. Suppose there are $N$ MAGs consistent with $\mathcal{P}$. For the complexity $T(N, d)$ of implementing Alg. 1 for $\mathcal{P}$ that contains $d$ vertices with circles, there is

$$T(N, d) = \sum_{i_1=1}^{k_1} T(N_{i_1}, d-1) + (2k_1 - 2)S,$$

where $N_{i_1}$ denotes the number of MAGs consistent with the PMG $\mathbb{M}_{i_1}$ that is obtained from $\mathcal{P}$ with the $i_1$-th local structures of $V_1$, $T(N_{i_1}, d-1)$ denotes the complexity of implementing Alg. 1 for $\mathbb{M}_{i_1}$.

We use an another search tree $\mathcal{T}_{outer}$ to denote the process of finding all $N$ MAGs, where each node in the $j$-th layer denotes a PMG which is obtained from $\mathcal{P}$ and local structures of $V_1, \cdots, V_{j-1}$, and the leaf node denotes a MAG consistent with $\mathcal{P}$. The format of the search tree is as Fig. 10. In the second layer ($j = 2$), there are $k_{i_1}$ PMGs due to $k_{i_1}$ sub-structures of $V_1$. We consider the $i_1$-th PMG $\mathbb{M}_{i_1}$ in the second layer. There are $N_{i_1}$ MAGs consistent with $\mathbb{M}_{i_1}$. Suppose there are $k_{2i_1}$ sub-structures of $V_2$ given a PMG $\mathbb{M}_{i_1}$. Using the calculation above, we can directly conclude that

$$T(N_{i_1}, d-1) = \sum_{i_2=1}^{k_{2i_1}} T(N_{i_1 i_2}, d-2) + (2k_{2i_1} - 2)S,$$

where $k_{2i_1}$ denotes the number of local structures of $V_2$ for the PMG in the $i_1$-th node on the second layer of $\mathcal{T}_{outer}$. Hence, there is

$$T(N, d) = \sum_{i_1=1}^{k_1} \sum_{i_2=1}^{k_{2i_1}} T(N_{i_1 i_2}, d-2) + 2S(k_1 + \sum_{i_1=1}^{k_1} k_{2i_1}) - 4S.$$

Repeat the above process, we could conclude that

$$T(N, d) \leq N + 2S(k_1 + \sum_{i_1=1}^{k_1} k_{2i_1} + \sum_{i_1=1}^{k_1} \sum_{i_2=1}^{k_{2i_1}} k_{3i_1 i_2} + \cdots) - 4S. \tag{1}$$

Note $k_1 + \sum_{i_1=1}^{k_1} k_{2i_1} + \sum_{i_1=1}^{k_1} \sum_{i_2=1}^{k_{2i_1}} k_{3i_1 i_2} + \cdots$ is essentially the number of edges in the search tree $\mathcal{T}_{outer}$. Hence, we only need to consider the number of edges in $\mathcal{T}_{outer}$ in the following.

Denote $Z$ the number of edges in $\mathcal{T}_{outer}$. Denote the number of node which has $j$ child nodes by $m_j$. Without loss of generality, suppose the maximum $j$ equals to $s$. As $Z$ equals to the number of edges in $\mathcal{T}_{outer}$, there are $Z - 1$ nodes in $\mathcal{T}_{outer}$. Due to the completeness of the orientation rules for incorporating local BK, there does not exist a node with only one child node. Hence $m_1 = 0$. There is

$$Z = m_1 + 2 * m_2 + \cdots + s * m_s$$
$$= 2 * m_2 + \cdots + s * m_s.$$

Consider the number of nodes, there is

$$Z - 1 = m_0 + m_2 + m_3 + \cdots + m_s.$$

With the two equations above, there is

$$m_0 = \sum_{j=2}^{s} (j-1)m_j - 1.$$

$m_0$ is essentially the number of leaf nodes, which equals to the number of MAGs consistent with $\mathcal{P}$. Hence,

$$\sum_{j=2}^{s} (j-1)m_j = N + 1.$$

Therefore, considering there must be at least one $m_j \neq 0, j \geq 2$,

$$Z = m_0 + m_2 + m_3 + \cdots + m_s + 1$$
$$\leq m_0 + \sum_{j=2}^{s} (j-1)m_j + 1$$
$$\leq 2N + 2.$$

Hence, according to Eq. 1, there is

$$T(N, d) \leq N + 2S(2N + 2) - 4S$$
$$T(N, d)/N \leq 1 + 4S.$$

Note $S$ denotes $\mathcal{O}(m^3 d^4)$, we conclude $T(N, d)/N \leq \mathcal{O}(m^3 d^4)$. □

## D.5. Proof of Proposition 2

*Proof.* We prove the soundness by contradiction.

For $\mathcal{R}_{17}$, suppose $A \leftarrow\!\!* B$ in $\mathcal{M}$ by $\mathcal{R}_1$. There must be $A \to D$ in $\mathcal{M}$. Due to the ancestral property and $D \twoheadrightarrow\!\!* C$, there can only be $D \to C$ in $\mathcal{M}$. Hence there is $A \to C$ due to the ancestral property and $C \to B$ in any MAG $\mathcal{M}$ consistent with the given PMG according to $\mathcal{R}_1$. Note in this case the sub-structure $A \to C \to B *\!\!\to A$ contradicts with the ancestral property. Hence $A \leftarrow\!\!* B$ is impossible.

For $\mathcal{R}_{18}$, suppose $A \to B$. As (a) $D$ is a possible ancestor of $T_1$ and $D$ is a possible ancestor of $T_2$, (b) $T_1$ is not adjacent to $T_2$ or an unbridged path relative to $\{T_1, T_2\}$, (c) $B \leftrightarrow D$, in any MAG consistent with the given PMG, there is either $D \to T_1$ or $D \to T_2$. Without loss of generality, suppose there is $D \to T_1$. There is $D \to T_1 \to \cdots \to A$. In this case $D$ is an ancestor of $B$ due to $A \to B$. However, there is $D \leftrightarrow B$, contradicting with the ancestral property. □

