# OpenReview forum: "Polynomial-Delay MAG Listing with Novel Locally Complete Orientation Rules"
_ICML.cc/2025/Conference — ICML 2025 oral_

### Official Review · Reviewer_gZDH · 2025-03-09

**Overall Recommendation:** 4

**Summary:**

This paper introduces an enhanced algorithm for the MAG listing task that outputs MAGs in the MEC with polynomial delay. Experimental results confirm the effectiveness of the proposed approach, and a counterexample construction is provided to demonstrate the incompleteness of current orientation rules.

**Claims And Evidence:**

Yes

**Essential References Not Discussed:**

I don't know any related works that are essential to understanding but are not currently cited/discussed in the paper.

**Experimental Designs Or Analyses:**

Given the theoretical contribution is solid in this paper, the experiments focus on verifying the effectiveness and efficiency of the proposed MAG listing algorithm on synthetic datasets. The experimental designs and results are sound and valid. Nevertheless, I would like to also suggest the authors add some discussions about real-world tasks and datasets.

**Methods And Evaluation Criteria:**

Yes

**Other Comments Or Suggestions:**

I do not have any other comments or suggestions.

**Other Strengths And Weaknesses:**

The paper is theoretically sound and solid and is well-written. My only concern is that it is unclear how the proposed algorithm could be used in practice. As the author mentioned in Sec. 1, causality is "a key component in numerous applications." Therefore, I think this paper could be stronger if there are some verifications on applications.

**Questions For Authors:**

I do not have other questions.

**Relation To Broader Scientific Literature:**

It may influence fields where the causality is curical with latent variables.

**Theoretical Claims:**

I looked through the theoretical claims and proofs, and do not find any issues.

---

> ### Author Rebuttal · Authors · 2025-03-31
>
> Thank you very much for your valuable comments and suggestions. We are grateful for your positive evaluation of the theoretical contributions and experimental designs in our paper. We also appreciate your suggestion to discuss real-world tasks and datasets, as well as to clarify the practical applications of the proposed MAG listing algorithm.
>
> 1.	How the algorithm can be used in practice:
>
> Below, we **outline three key applications** where the proposed algorithm can be used:
>
> (a). Intervention Variable Selection:
>
> In causal discovery with active interventional data, selecting the optimal variables for intervention can significantly reduce the number of required interventions, which is critical given the high cost of interventions in real-world scenarios. In the causal sufficiency setting, He & Geng (2008) proposed a maximum entropy criterion for selecting variables based on the distribution of local structures in the Markov equivalence class (MEC) of DAGs. This requires efficient DAG listing to calculate the frequencies of local structures. In the causal insufficiency setting, MAG listing can generalize this approach by enabling the computation of analogous distributions for MAGs, allowing for optimal intervention variable selection.
>
> (b). Determining the Distribution of Possible Causal Effects Given a Markov Equivalence Class (MEC):
>
> Given a MEC learned from observational data, sometimes we want to estimate the distribution of possible causal effects of one variable $X$ on another $Y$. In this case, it is necessary to consider all possible causal graphs within the MEC. Note different causal effects may appear with different possibilities since they may correspond to different numbers of causal graphs. In the causal sufficiency setting, Maathuis et al. (2009) enumerated all DAGs in an MEC to determine the distribution of possible causal effects. In the causal insufficiency setting, MAG listing method allows for determining the distribution of possible causal effects.
>
> (c). Complete Causal Discovery Algorithms:
>
> In causal discovery tasks, completeness is a desirable property, ensuring that all valid causal graphs (DAGs or MAGs) consistent with the data are considered. For example, in Appendix D.4 of Kocaoglu (2023) and Alg. 1 of Gerhardus (2024), MAG listing is explicitly used to achieve completeness in causal discovery. Efficient MAG listing allows these algorithms to systematically explore all equivalent MAGs, ensuring that no potential causal graphs are overlooked.
>
> In summary, the three applications-intervention variable selection, determining distributions of possible causal effects, and enabling complete causal discovery algorithms—highlight the importance of MAG listing in advancing causal inference methods under causal insufficiency. The development of efficient MAG-listing algorithms makes these applications feasible and opens new possibilities for research and practical implementation.
>
> In the revised version, we will elaborate on the applications of MAG listing.
>
> 2.	Discussion about datasets:
>
> In our experiments, we focused on synthetic datasets because they allow us to systematically evaluate the correctness, effectiveness, and efficiency of the proposed algorithm under controlled conditions. Synthetic data also enable us to test the algorithm across a wide range of parameter settings, which would be difficult to achieve with real-world datasets.
>
> To the best of our knowledge, there are currently no publicly available datasets with ground-truth causal graphs that account for the causal insufficiency setting (i.e., the presence of latent confounders and selection variables). This lack of ground-truth data makes it difficult to directly validate the algorithm on real-world datasets in this context.
>
> We fully agree that testing the algorithm on real-world datasets would provide stronger evidence of its practical utility. As part of future work, we plan to explore real-world datasets that approximate the causal insufficiency setting. Additionally, we hope that the community will develop and release open datasets containing both latent confounders and selection variables, which would further advance research in this area. We will explore the possibility of incorporating real-world case studies in future work to further strengthen the practical contributions of this research.
>
>
> **Beyond MAG listing, we want to highlight the contributions of our proposed orientation rules. In causal insufficiency setting, the analogical ``Meek rules`` for incorporating BK into a MEC have remained an open problem for many years. We believe our new rules make a significant advancement towards addressing this fundamental problem.**
>
> Thank you once again for your thoughtful feedback, which has helped us improve the clarity and impact of our work.

---

> > ### Comment · Reviewer_gZDH · 2025-04-01
> >
> > I appreciate the detailed response. I fully agree that there should be many potential applications, and I agree that testing on synthetic datasets has many advantages.
> >
> > > To the best of our knowledge, there are currently no publicly available datasets with ground-truth causal graphs that account for the causal insufficiency setting (i.e., the presence of latent confounders and selection variables). This lack of ground-truth data makes it difficult to directly validate the algorithm on real-world datasets in this context.
> >
> > To my knowledge, there are some papers that consider the settings where some variables from real benchmarks are hidden. It seems to come from the definitions of the ancestral graphs by [1]. It serves as a simulation of the causal insufficiency scenario. I am just curious whether this setting could be extended into your experiments.
> >
> > [1] Thomas Richardson, Peter Spirtes. Ancestral graph Markov models. 2022

---

> > > ### Author Response · Authors · 2025-04-04
> > >
> > > Thank you for providing this insightful idea. Simulating scenarios with latent confounders and selection bias by hiding certain variables from real benchmarks, as inspired by the definitions of ancestral graphs in [1], is a practical approach. Also, we greatly appreciate your suggestions regarding the verification of real-world applications, which are instrumental in improving our work and enhancing its impact.
> > >
> > > In the revised version, we will incorporate the application of MAG listing for interventional variable selection and conduct a more detailed empirical study on intervention numbers in the presence of latent variables to uncover all causal relationships.
> > >
> > > Following your suggestion, we conducted a preliminary experiment using real-world data with observational and interventional data from Sachs et al. ([2]). The dataset processed from [3] consists of 7466 measurements of the abundance of phosphoproteins and phospholipids recorded under different experimental conditions in primary human immune system cells. The eleven features include Raf, Mek, PLCg, PIP2, PIP3, Erk, Akt, PKA, PKC, p38, JNK. In addition to observational data, the dataset also includes interventional data, with interventional targets being Akt, PKC, PIP2, Mek, and PIP3, respectively.
> > >
> > > In this experiment, we focus on the task of causal discovery and apply our MAG listing method to select interventional variables using the maximal entropy criterion. We randomly select three variables as latent variables. We compared two strategies for selecting interventional variables: the maximal entropy criterion and the random strategy.
> > >
> > > For the random strategy, an observed variable with circles is randomly selected for intervention.
> > >
> > > For the maximal entropy criterion, the interventional variable $V$ is selected based on maximizing the formula:
> > >
> > > $H_V=\sum_{i=1}^{M}\frac{l_i}{L}log(\frac{l_i}{L})$,
> > >
> > > where $l_i$ denotes the number of MAGs with $i$-th local structure of $V$ in the MEC, and $L$ is the total number of MAGs. To calculate the number of MAGs for each local structure, we first list all the MAGs by our method.
> > >
> > > We record the total intervention number for each criterion. Note that the experiments are not fully realistic, as some interventional variables lack corresponding interventional data. We directly introduced the true non-circle marks for the simulated intervened variables.
> > >
> > > We conducted ten simulations, randomly selecting three latent variables in each run. The experimental results are summarized as follows.
> > >
> > > | Criterion                | Total Intervention numbers(10 simulations)|
> > > |--------------------------|--------------------|
> > > | Maximal Entropy Criterion | 41                |
> > > | Random Criterion          | 50                |
> > >
> > > The results demonstrate that selecting the interventional variable by maximal entropy criterion is effective, which highlights the application of MAG listing algorithm.
> > >
> > > Thank you once again for your thoughtful feedback and valuable input, which is significantly helpful for improving our work. We will conduct further empirical studies to verify potential real-world applications, highlighting the necessity and utility of MAG listing.
> > >
> > > [2] K. Sachs, O. Perez, D. Pe’er, D. A. Lauffenburger and G. P. Nolan. Causal protein-signaling networks derived from multiparameter single-cell data. Science 308.5721 (2005): 523-529.
> > >
> > > [3] Wang, Y., Solus, L., Yang, K., and Uhler, C. Permutationbased causal inference algorithms with interventions. In Advances in Neural Information Processing Systems, pp. 5824–5833, 2017.

---

### Official Review · Reviewer_Wkax · 2025-03-10

**Overall Recommendation:** 4

**Summary:**

The paper presents a novel polynomial-delay algorithm for listing all maximal ancestral graphs (MAGs) in a Markov equivalence class (MEC) while incorporating singleton background knowledge (BK). The core contribution is the development of three new orientation rules that improve computational efficiency compared to existing methods. The authors provide formal proofs of soundness and local completeness of the proposed rules and compare their algorithm’s performance with previous approaches through experiments on synthetic data. They also discuss the limitations of existing methods and the incompleteness of current orientation rules for incorporating general BK, motivating future research.

**Claims And Evidence:**

- The claims of polynomial-delay of the proposed algorithm and soundness and locally completeness of the proposed orientation rules for incorporating singleton BK are supported by theoretical proofs.
- The claim that MAGLIST-POLY significantly outperforms previous methods in efficiency is supported by experiments.

**Essential References Not Discussed:**

Not clear due to limited familiarity. But the discussion w.r.t related work is smooth and clear.

**Experimental Designs Or Analyses:**

The authors generate random ER graphs, convert them into MAGs, and compare listing algorithms under different densities and graph sizes. They use 100 random graphs per setting, with time limits of 3600 seconds per experiment. The design and conclusion make sense to me.

**Methods And Evaluation Criteria:**

- The paper builds on previous MAG listing algorithms but improves efficiency by introducing singleton BK and new orientation rules.
- The method is evaluated by correctness, computational complexity and experimental validation.

**Other Comments Or Suggestions:**

- Apart from ER graphs in the experiments, it would also be helpful to consider scale-free graphs to reflect real-world causal structures.

**Other Strengths And Weaknesses:**

**Strengths**
- It provides the first polynomial-delay MAG listing algorithm.
- It proves the soundness and local completeness of orientation rules.
- The experiments show noticeable improvement.

**Weaknesses**:
- Due to the theoretical nature, the paper is dense in mathematical notations and concepts and not easy to comprehend for average readers.

**Questions For Authors:**

- It there any intuition or idea how to further optimize the rule applications to reduce the $O(m^5d^4)$ complexity?
- In the experiment for $d=6$, why the computation time has a local drop for $\rho=0.15$?
- How does the "brute-force" in the experiments work?

**Relation To Broader Scientific Literature:**

Structure learning is widely used in scientific research where latent variables as confounders are pervasive. Using MAG for flexble modelling and understanding the possible Markov equivalent graphs are essential to the application of these methods.

**Theoretical Claims:**

The main theoretical contributions include:
- Theorem 1: The new orientation rules are locally complete for incorporating singleton BK.
- Theorem 2: The algorithm outputs all and only the MAGs in the MEC.
- Theorem 3: The method has polynomial delay $O(m^5d^4)$ where $m$ is the numebr of edges.

The proofs appear rigorous, leveraging formal graph-theoretic arguments, while not carefully checked.

---

> ### Author Rebuttal · Authors · 2025-03-31
>
> Thank you for insightful comments and positive evaluation! We will add more illustrations and examples to make it easier to read for average readers in the revised version.
>
> 1. Is there any intuition or idea how to further optimize the rule applications to reduce the complexity:
>
> Thank you for raising this insightful question. In the submitted version, we mainly focused on the polynomial-delay and the orientation rules to incorporate BK, without delving deeply into the specific complexity of rule application. Your question, along with Reviewer ZQmR’s comments, prompted us to carefully examine the complexity, and we found that it can indeed be improved to $O(m^3d^5)$. Below, we provide a rough analysis of the implementation of $R_{14}$, which is the most time-consuming part.
>
> There are $d$ variables. For each variable $X$, there are $ O(d^2)$ pairs of $V_i,
> V_j$ such that $V_i\ast-\circ X\circ-\ast V_j$, and detecting whether $V_i$ is prior to $V_j$ (or vice versa) relative to $X$ takes $O(m^3d^2)$. Additionally, there are $O(d)$ variables $V$ such that $X\circ\rightarrow V$ is possibly oriented by $R_{14}$. For each such $V$, since we already know which variables are prior to $V$ relative to $X$, enumerating each pair $T_1,T_2$ prior to $V$ takes $O(d^2)$, and detecting conditions (a) and (b) in $R_{14}$ takes $O(m)$. Hence, the total complexity is $d* (O(d^2)* O(m^3d^2)+ O(d)* O(d^2)* O(m))=O(m^3d^5)$.
>
> The main computational bottleneck lies in detecting the "prior to" relation defined in Def. 3. Since we check this relation for every pair, the computational cost is high. In our current analysis, we did not exploit certain properties of the "prior to" relation that could potentially reduce the number of pairs to consider or improve the detection efficiency. Exploring these properties could lead to further optimization, which we leave for future work.
>
> 2. In the experiment for $d=6$, why the computation time has a local drop for $\rho=0.15$:
>
> Thank you for your careful observation. This is an excellent and thoughtful question. We have analyzed the phenomenon and found that it is not caused by the algorithm itself but rather by the **parallel implementation** of the simulations.
>
> Specifically, we have ever tried to run simulations for $d=6$ under different parameter settings ($\rho=0.1$ and $\rho=0.15$) **separately**, the average running time for $\rho=0.15$ is indeed higher than $\rho=0.1$. However, in our experiments, we adopted the same parallel implementation manner as Wang (2024a), where each thread processes different combinations of $d\in${6,8,10,12,14},$\rho\in${0.05,0.1,0.15,0.2,0.25},$graphindex\in$[1:100]. Under this parallel setup, although the method returns **totally same** MAGs as the method implemented under the separate setup for each parameter, the running time for $d=6,\rho=0.1$ and $d=6,\rho=0.15$ can exhibit slight variations. Hence we analyze it is due to the differences in resource allocation across threads. For $d=6$, the computation time for each PAG is extremely short (ranging from 0.000x seconds to 0.0x seconds, with some recorded as 0). In such cases, the dominant factors influencing running time are not the algorithm itself but external factors such as thread-level resource allocation. When $d$ and $\rho$ are relatively large, the algorithm’s implementation becomes the primary time-consuming part, and this anomaly no longer occurs.
>
> In the revised version, we will include an explanation to clarify this phenomenon. Thank you!
>
> 3.	How does the "brute-force" in the experiments work:
>
> Thank you for pointing out this issue—it is indeed a result of insufficient clarity in our writing. The brute-force method used in our experiments follows Alg. 4 of Wang (2024a). Below is a brief explanation of its main idea:
>
> (a)	Enumerating configurations: Given a PAG, the method enumerates all non-circle configurations of all circles (Line 5 in Alg. 4 of Wang (2024a)). This process generates multiple mixed graphs.
>
> (b)	Filtering Mixed Graphs: For each mixed graph, the algorithm checks whether it satisfies three conditions: ancestral property, maximal property, and consistency with the given PAG (Line 6-12 in Alg. 4 of Wang (2024a)). If the mixed graph meets these criteria, it is included in the output (Line 13 in Alg. 4).
>
> In the revised version, we will add the brute-force algorithm in the appendix for reference and provide a concise introduction in the main paper to avoid confusion.
>
> **Beyond MAG listing, we want to highlight the contributions of our proposed orientation rules. In causal insufficiency setting, the analogical ``Meek rules`` for incorporating BK into a MEC have remained an open problem for many years. We believe our new rules make a significant advancement towards addressing this fundamental problem.**
>
> We sincerely appreciate your thoughtful feedback and hope these clarifications address your concerns. We welcome any additional questions.

---

> > ### Comment · Reviewer_Wkax · 2025-04-03
> >
> > I thank the authors for their detailed response, which addresses most of my questions. I keep my score unchanged.

---

### Official Review · Reviewer_dgb7 · 2025-03-10

**Overall Recommendation:** 5

**Summary:**

This paper proposes a MAG listing algorithm (i.e., output all and only MAGs in the MEC, represented by a PAG) with polynomial delay.

---

## update after rebuttal:

I thank the authors for the clear explanation to my questions. I keep my score of acceptance.

**Claims And Evidence:**

Yes.

**Essential References Not Discussed:**

I would suggest authors to also discuss the DAG listing algorithms. To the best of my knowledge their current best time complexity is O(n^4) where n is the number of nodes (https://arxiv.org/abs/2301.12212).

Since DAG is a specific kind a MAG, it would be interesting to see how the technical development of DAG listing algorithm share some same insights that can benefit MAG listing, or which technical obstacle is unique to the MAG case.

**Experimental Designs Or Analyses:**

Yes.

**Methods And Evaluation Criteria:**

Yes.

**Other Comments Or Suggestions:**

/

**Other Strengths And Weaknesses:**

This work is novel and crucial. I do not see major weaknesses.

**Questions For Authors:**

1. If the purpose is not enumerating all the MAG instances in the MEC but just to count the size of the MEC, would the time complexity be different? Would the proposed method in this paper still help?

2. If one knows a-priori that there is no selection bias (so that there is no -- edge and all -o edge can be oriented to ->), or similarly no latent variables, can this prior knowledge also be incorporated into the MAG listing algorithm? Will the rules still be sound and locally complete? Will the time complexity be different?

**Relation To Broader Scientific Literature:**

To me, the result presented in this paper is very necessary and crucial for the field of causal discovery.

Several problems that can directly benefit from this work include FCI with background knowledge, causal effect estimation from PAG, and experimental design with latent variables.

**Theoretical Claims:**

Yes I read the theorems, ran the examples, and skimmed through the proofs. They look correct to me. But I cannot guarantee.

---

> ### Author Rebuttal · Authors · 2025-03-31
>
> We sincerely appreciate your positive evaluation! We will incorporate the discussion you mentioned (DAG listing, counting…) in the revised version.
>
> 1. …to count size… would complexity be different? Would the method help:
>
> Thank you for your insightful question. Currently, we do not have a clear idea about MAG counting under latent confounding, although DAG counting is thoroughly solved (Wienöbst, 2021). The two main challenges are: (1) bi-directed edges prevent using an order to describe uncertain edge orientations, and it is unclear what mathematical tool could replace order in this context; (2) the orientation of $\circ\rightarrow$ connecting different circle components can influence the orientation within a circle component. Consider $A\circ-\circ B\circ\rightarrow C\leftarrow\circ A$. If there is $B\leftrightarrow C\leftarrow A$, there must be $A\ast\rightarrow B$, showing that the orientations within a component are not independent of the outside orientations. Due to these challenges, we do not have a clear idea about MAG counting method which does not need MAG listing. However, in cases with **only** selection variables, the two obstacles might be resolved: (1) since there cannot be an edge into an undirected edge and no bi-directed edges, an order could be defined among variables without undirected edges; (2), all edges connecting components are directed, thus avoiding the scenarios above.
>
> If the two obstacles could be solved, we conjecture that some results, such as Lemma 2 and locally complete rules, may be helpful. Nevertheless, without addressing these obstacles, we are unable to provide a definitive answer at this time.
>
> 2. …a-priori no selection bias or no latent variables, can this prior knowledge be incorporated? Will the rules be sound and locally complete? Will the time complexity different:
>
> Thank you for this excellent question. Yes, **this prior knowledge can be incorporated, but it requires an additional rule: orient $-\circ$ to $\rightarrow$ (no selection variable) or orient $\circ\rightarrow$ to $\rightarrow$ (no latent confounding)**. To ensure Thm. 2 holds in the new setting, two key results must be proven: (A) **the new set of rules are sound and complete to incorporate a valid local transformation (LT)**. This is not required in our paper because Wang 2024a has proven that a subset of our proposed rules is sound and complete. However, under the new setting, this must be proven; (B) **the new set of rules are sound and locally complete for singleton BK.** Soundness is evident. We only prove locally completeness. (A) ensures the validity of outer loop of Alg.1 (Line 322-324), (B) ensures that of inner loop (324-329). Due to space limit, we only give a proof sketch.
>
> No-selection bias: for (A), we refer to Thm. 2 of Wang (2023). Note the PAG in Thm.2 can be easily generalized to a PMG compatible with LT. For (B), following Wang (2023), PMG compatible with LT contains no $-\circ$. We will prove that Lemmas 3,4 in our paper still hold. Lemma 3 evidently holds. For Lemma 4, consider $H$ on Line 832. If $H$ contains no $\circ-$, then we directly follow the proof of Lemma 4. If $H$ contains $\circ-$, it can be shown that the singleton BK transforms at least one edge $X\circ-\circ V$ to $X-\circ V$. In this case, there cannot be an edge $V’\ast\rightarrow X$, as this would force $X-\circ V$ to $X\rightarrow V$ by $R_{16}$. Thus $\mathbf{C}$ on Line 830 is empty. Thus, if $H$ contains $X\circ-\ast T$, it can be shown that $\mathbf{C}’={T}$ satisfies the four conditions in Lemma 2 as it contains only one element, ensuring a MAG consistent with $H$ and $X\leftarrow\ast T$. Lemma 4 holds.
>
> No-latent confounding: for (A), consider a PMG $M$ compatible with LT and no $\circ\rightarrow$. It suffices to prove that no bi-directed edges are formed by Alg.2 of Wang (2024a), which would otherwise lead to invalid structure $\leftrightarrow$.
>
> According to Lemma 22 of Wang (2024a), bi-directed edges can only form in Step 2 of Alg.2. Note $M$ contains no $\circ\rightarrow$. If $K\circ\rightarrow T$ in Step 2 is transformed to $K\leftrightarrow T$, according to balance property and $K\in$PossDe\Z, there is $X\circ\rightarrow T$ in $M$, contradiction. We can thus directly apply Thm. 2 and Thm. 3 of Wang (2024a) to prove (A). For (B), the proof follows almost entirely from our paper, as Lemma 3 and 4 unaffected by the new rule.
>
> **The complexity changes without latent confounding.** In this case, there are no unbridged paths (Def.1), as their existence would imply a bi-directed edge. Hence, detecting the third condition in Def. 3 ($O(m^3d)$) is unnecessary, leaving only $O(m)$ for the first two conditions. Thus the complexity of $R_{14}$ improves by $O(m^2d)$. Similarly, all rules involving unbridged paths can be applied more efficiently. However, for no selection bias, the complexity of $R_{14}$ remains unchanged.
>
> We appreciate your positive and thoughtful feedback, and welcome additional questions.

---

### Official Review · Reviewer_ZQmR · 2025-03-15

**Overall Recommendation:** 4

**Summary:**

The paper proposes a method for enumerating (listing) MAGs consistent with a given PAG with a polynomial delay, i.e., the ratio between the time complexity of enumeration and the number of consistent MAGs is polynomial in the graph size. The method is based on resolving one circle at a time, followed by applying orientation rules to direct as many edges as possible. In particular, the paper introduces three novel orientation rules, which, along with existing rules, are sound and locally complete for incorporating singleton background knowledge (BK). These rules ensure that the listing algorithm has a polynomial delay.

**Claims And Evidence:**

Proofs for theorems were included in the appendix. The setup for experiments is clear.

However, I found some of the proofs lengthy and difficult to follow (e.g., Lemma 3 & 4). I highly suggest authors provide sketches and intuitions/examples for each part of the proof -- maybe break them into smaller lemmas.

**Essential References Not Discussed:**

I'm not aware of any essential references not discussed.

**Experimental Designs Or Analyses:**

I checked the experiments section and the results made sense to me. The listing algorithm proposed in the paper was more efficient than bruteforce and the previous method in [Wang, 2024a] with exponential delay.

**Methods And Evaluation Criteria:**

The paper contains both theoretical results and simulations The method is based on proposing more orientation rules that are sound and locally complete for incorporating BK, which can be exploited to accelerate the listing of MAGs.

**Other Comments Or Suggestions:**

- pg 3: Definition 1 is ill-defined. This notion is used frequently later so please make sure it's defined accurately. Also, please cite it if the notion was introduced previously in [Wang, 2024]. I would suggest moving "Intuitively, given ... must be an ancestor of V'" after the Definition.

- pg 4, col1, line 200 "fig. 1(d)" -> "fig 1(c)"

- pg4, col1, line 214-216 "Since the rules ... due to $A \circ\rightarrow B$" confusing.

- pg4, end of section 3.1. Please elaborate more on how the enumeration of each circle can be viewed as a type of background knowledge (BK). I don't think this connection is well articulated in the paper. If possible, I'd suggest providing some concrete examples.

- pg 5, col 1, line 230. It took me a while to figure out what $R_{12}$ is. Please make sure you mention that Rules 1-13 are shown in the Appendix.

- pg 5 lines 261-267. I don't think the intuition here is explicit enough. It's still hard for me to understand what's special about the "prior to" relation in Definition 3. Is it possible to explain it more succinctly?

- pg 5 line 273: two "an"'s.

- pg 7 col 2 lines 376-379. Mention the examples are shown in Figure 7.

**Other Strengths And Weaknesses:**

Strengths:
- The topic of listing MAGs from PAGs is important, and the results (assuming correct) are significant since they constitute the first polynomial listing methods on MAGs.
- The paper contains both theoretical results and experiments to demonstrate the improvement.

Weaknesses:
- The main results rely on complicated graphical notions without clear intuitions, making them somewhat difficult to follow. The main Definition 1 contains typos. See more details below.

**Questions For Authors:**

- Unless I missed something, I don't think all local transforms produce a PMG that has a consistent MAG. In particular, how do you check if $H'$ obtained from $H$ (Algorithm 1 lines 20, 22) always has a valid MAG? Consider Figure 7(a), for example, suppose we have run LocalTransform on variable D to set edges $D - \circ B$ and $D - \circ C$. Now if we run LocalTransform on $C$, we will try (i) $C \leftarrow \circ A$; and (ii) $ C - \circ A$ for the edge $C \circ - \circ A$. However, case (i) here will yield a PMG that is not consistent with any MAG (by $R_{17}$), which brings additional cost to the search since it cannot lead to any MAG. Are there any methods to prevent such cases? Also, was this issue addressed in the Proof of Theorem 3 (Appendix D.4)?

- I'm curious about the succinctness of the new orientation rules proposed in this work. Do you believe there may be simpler rules, which, combined with other existing rules, can imply some of the complicated rules (e.g., $R_{14}, R_{18}$) in this paper?

- Could you please provide some specific applications for MAG listing?

**Relation To Broader Scientific Literature:**

The paper improves upon the previous work in [Wang et al, 2024] by introducing additional orientation rules that empower the polynomial listing of MAGs.

**Theoretical Claims:**

I checked the proofs for Proposition 1, Theorem 2, Theorem 3, and Proposition 2.

---

> ### Author Rebuttal · Authors · 2025-03-31
>
> Thank you for careful reading and providing many valuable and constructive suggestions, which will definitely help us improve our work. As suggested, we will provide proof sketches and examples for Lemmas 3/4 in the revised version.
>
> 1.	…I don't think all local transforms produce a PMG that has a consistent MAG…:
>
> Thank you for raising this critical question. We can confirm that the cases you described will never occur in our proposed algorithm. Below, we will explain (a) why the example you mentioned has been prevented, and (b) why cases where all local transformations produce a PMG not having a consistent MAG will not happen in our algorithm.
>
> (a)	According to Alg.1, when we locally transform $D\circ-\circ C$ to $D-\circ C$ given the PMG in Fig.7(a), the orientation rules are applied immediately after the local transformation (Line 20,22 of Alg.1). As a result, $R_7$ orients $C-\circ A$, ensuring that $C\leftarrow\circ A$ is not considered in subsequent transformations.
>
> Further, note in Function LocalTransform(H,X), all circles at X are recursively transformed. Although transforming $D\circ-\circ B$ to $D-\circ B$ **itself** cannot orient $C-\ast A$ by $R_1-R_{16}$, $C-\ast A$ can still be always obtained by $R_7$ or $[R_{16},R_{13}]$ **after the following transformation** of $D\circ-\circ C$ to $D-\circ C$ or $D\leftarrow\circ C$. Hence the problematic structure $C\circ-\ast A$ will not occur during subsequent transformations on $C$.
>
> (b)	The examples in Fig.7 imply the incompleteness of the rules. However, in LocalTransform(H,X) in Alg.1, we **only** transform the circle at $X$, followed by applying rules. The locally complete property ensures that transforming the circles at $X$ will not result in a PMG without a consistent MAG. Once all circles at $X$ are transformed into non-circles, the resulting non-circles at $X$ **essentially correspond to a valid local transformation** proposed by Wang (2024a), as detailed on Line 984. Due to two facts: (1) Wang (2024a) proved that $R_1-R_{10}$ as well as the first case of $R_{16}$ are sound and complete to incorporate valid local transformations given a PMG compatible with local transformations, (2) our rules are sound and properly contain the rules of Wang (2024a), we conclude that **the PMG obtained after transforming all circles at $X$ into non-circles and applying our rules (Line 17 of Alg.1) is equivalent to the PMG obtained by applying the local transformations and rules of Wang (2024a)**, as detailed on Line 992. Wang (2024a) proved that such PMGs are compatible with local transformations and fulfill complete property. Thus, local transformations on PMGs obtained on Line 17 of Alg.1 will not yield a PMG without a consistent MAG.
>
> 2.	The succinctness of rules:
>
> Thank you for this excellent question. We discuss it from two perspectives: (1) whether a simpler rule combined with the existing rules can replace $R_{14}/R_{18}$; (2) whether the rules themselves can be simplified.
>
> For (1), we believe that such a simpler rule is unlikely to exist. Suppose, for contradiction, a simpler rule combined with other existing rules can replace $R_{14}/R_{18}$. Then, in any cases where $R_{14}/R_{18}$ is triggered, **the existing rules alone should transform at least one circle**. However, in the examples in Fig.4(c) and Fig.7(d), when adding BK $C_1\leftrightarrow X\leftrightarrow C_2$ and $D\leftrightarrow B$ respectively, no existing rules except for $R_{14}/R_{18}$, can transform any circles. The examples suggest that the existing rules play no role, contradiction. Thus no simpler rule, combined with existing rules, can replace $R_{14}/R_{18}$.
>
> For (2), if the goal is to simplify the rules themselves, we think it is hard to achieve but cannot rule out the possibility. However, we would like to highlight the contributions of our proposed rules. In causal insufficiency, finding an analog of ``Meek rules`` for incorporating BK into a MEC is an open problem for many years. Our rules can orient edges that existing rules fail to handle. We believe these rules make a significant advancement toward addressing this fundamental problem.
>
> 3. MAG listing application:
>
> Thank you for your question. Due to space limits, we outline three applications in Response 1 to Reviewer gZDH. We will incorporate them in our revised version.
>
> 4. It's hard to understand what's special about "prior to" relation in Def.3:
>
> Thank you for raising this question. Intuitively, consider a sub-structure $A\ast-\circ X\circ-\ast B$ in $H$, the relation “$A$ is prior to $B$ relative to $X$’’ characterizes the case where $X\leftarrow\ast A$ must be oriented given the transformation of $X\circ-\ast B $ to $X\leftarrow\ast B$. The three conditions in Def. 3 are three possible cases for that. We will adjust the expression to make it clearer in the revised version.
>
> We sincerely appreciate your thoughtful feedback and hope these clarifications address your concerns. We welcome any additional questions.

---

### Decision · Program_Chairs · 2025-05-01

**Decision:**

Accept (oral)

**Comment:**

The authors consider the problem of listing MAGs from a PAG. In causal discovery, MAG represents a class of causal graphs with latents (and selection bias) that all entail the same conditional independence relations. PAG represents an equivalence class of MAGs. PAG is what is learnable from data. MAG listing is a very hard problem, and the presented paper provides the first polynomial delay algorithm to do this. It fills an important fundamental gap in our understanding of observational causal discovery. Their approach relies on replacing a critical step of the previous algorithm from jointly orienting all circle marks locally to orienting them one at a time. This basic idea can only be realized properly if complemented with a sound and complete set of orientation rules. The authors develop these in this paper. They also show that even after we add the orientation rules in this paper, we still do not have global orientation completeness. This is perhaps not surprising, but it is also very valuable to establish formally through counterexamples.

The reviews have been positive, and after rebuttal, all reviewers recommend acceptance, acknowledging that it fills an important gap in the literature. The only minor weakness is seen as dense notation and dense theory. But I think this is unavoidable for a paper like this.

The notion of local completeness is interesting and valuable, in my opinion. However, its necessity in the context of the paper is a bit less clear. The authors argue that without it, we would have a problem because we would potentially create orientations that are inconsistent with the Markov equivalence class. However, it is unclear why this is such a big issue. We can check if the current partial structure is still consistent with the Markov equivalence class learned from data. This would incur computational overhead, but it is not obvious that this is a deal-breaker, in the sense that such an algorithm would not be polynomial time. Since the authors mention the main reason the previous method had exponential time was that it jointly oriented all circles at one node. Therefore, it is important for the authors to either provide a counterexample that, without local completeness, their algorithm would not be polynomial time in the worst case, or refrain from suggesting the necessity of local completeness as a precondition for poly delay. As I said, local completeness is already valuable on its own, in my opinion.